# Below the Reliability Floor: Recovering True Success from Judge-Gated Loops

## Abstract

LLM judges are increasingly placed *inside* an agent's loop, scoring the agent's own attempts and re-prompting until one passes. We show this quietly corrupts measurement: retry-until-PASS is *optional stopping* against a noisy classifier—it keeps drawing until the judge slips—so the reported pass rate is a biased estimator of true success, upward in the pass-prone retry regimes of interest (and downward under conservative rules such as strict rubrics or unanimous juries). We make this exact. The cap-$K$ gate is a binary classifier with closed-form sensitivity/specificity, and its bias is governed by one coefficient, the gate's Youden index $J$: as $J \to 0$ the gated rate becomes uninformative about $\pi$, so recovery must fall back to gold labels and no estimator beats the gold-only mean. Across 44 capable-agent loops on GSM8K, MATH, and code with objective ground truth (no authored weakness; a separate terse-agent stress set is excluded here) the inflation is systematic (median slip +0.16; worst on code, where the judge cannot run the candidate, a true 0.74 inflated to a reported 0.98) and obeys a closed-form law predicting the slip from per-attempt statistics (pooled $r=0.95$; errors-in-variables slope 0.765 $[0.70, 0.84]$, excluding 1). To recover true success we benchmark Rogan–Gladen against prediction-powered inference: **the label-efficient estimators (PPI/PPI++) recover $\pi$ far more accurately than naive reporting and Rogan–Gladen** (mean recovery MAE 0.050 vs. 0.149, and 0.081 vs. 0.241 as gold becomes scarce), because they escape the $1/J^2$ variance that makes the classical correction fragile; on balanced low-bias gates they match the gold-only mean (per-gate differences within bootstrap noise, none uniformly best), so our contribution is not a uniquely-best estimator but the identification of loop-induced bias and the $J$ diagnostic that says when—and whether—to correct at all. Beyond verifiable gold, we *measure* recovery on public, human-labeled *non-verifiable* gates—response safety and summary quality—where PPI++ recovers the true rate to mean MAE 0.043 versus naive 0.165 ($\sim 4\times$ in aggregate, up to $> 10\times$ on the most-biased gates); and—in the motivating regime, a non-verifiable safety judge *inside* a real retry loop ($n=400$, against a pre-registered 3-model strong-LLM panel—a disclosed proxy, human-anchored at raw 0.90 agreement)—a lenient gate ships 6.8% *panel-labeled*-unsafe responses (95% CI $[0.047, 0.096]$) while the calibrated correction recovers the panel safe-rate $\sim 3.5\times$ more accurately than naive. The deliverable is a recipe: report a PPI++ estimate alongside $J$ as a reliability/identifiability diagnostic, measure at $K=1$, and—via a label-free drift detector (ROC-AUC 0.80)—de-bias only when calibration transfers. We release all code and content-free data.

## 1 Introduction

A growing class of agentic systems puts a language-model judge in the control loop. The agent produces work; an LLM judge scores it; on a FAIL the agent revises or re-runs and is judged again; the loop stops when the judge returns PASS (or a retry budget is exhausted). The fraction of items that end in PASS is then reported as the system's *success rate*—the headline number on dashboards, in model cards, and in procurement decisions.

This number is a measurement, and the conditions under which it is a *valid* measurement have not been examined. The judge is not a fixed instrument: re-running the same judge on the same item at temperature zero flips its verdict at a non-trivial rate (Feuer et al., 2025; Guerdan et al., 2025), and that noise concentrates on the hard, consequential cases. Crucially, the loop does not sample the judge once—it samples *until it gets a PASS*. Retrying never turns a PASS into a FAIL; it can only convert a FAIL into a PASS. So any item the judge would pass on *some* run eventually ships, including a genuine violation the judge catches only intermittently. This is optional stopping against a noisy classifier: like repeated significance testing on accumulating data (Armitage et al., 1969) or reusing a held-out evaluator until it agrees (Dwork et al., 2015; Berk et al., 2013), the stopping rule biases the reported quantity even though no individual judgment is dishonest.

We treat this as a measurement-science problem and contribute a constructive fix rather than only a diagnosis. The estimator we use is classical—our contribution is not a new statistic but the identification of loop-gated success as a prevalence-under-imperfect-test problem (equivalently, the *quantification* / Adjusted Classify-and-Count task), the result that retrying actively erodes the gate's discrimination in the pass-prone regime (so more retries means a worse measurement there), and the resulting operational recipe. Our contributions:

- **The gate is a classifier (§2).** We show a cap-$K$ retry-until-PASS gate is a binary classifier of true compliance with closed-form, $K$-dependent sensitivity $\mathrm{Se}(K)$ and specificity $\mathrm{Sp}(K)$, so the reported success rate $R(K) = \pi\mathrm{Se}(K) + (1 - \pi)(1 - \mathrm{Sp}(K))$ is a biased estimator of true success $\pi$.

- **Retrying erodes discrimination (§3).** The gate's Youden index $J(K) = \mathrm{Se}(K) + \mathrm{Sp}(K) - 1$ is non-increasing in $K$ under a pass-prone stochastic-dominance condition (Prop. 1)—because retries add false passes on violations faster than they rescue true passes on already-passing compliant work—and always converges to a limit $J(\infty)$. The operative *reliability floor* is the identifiability boundary $J \to 0$, where the gated rate stops carrying information about $\pi$; outside the pass-prone regime $J$ can instead *rise* with $K$.

- **Recovery, identifiability, and the right estimator (§4).** We recover $\pi$ by correcting for the gate's class-conditional rates from a small gold-labeled calibration set, and benchmark estimators head-to-head: *PPI/PPI++* (Angelopoulos et al., 2023a) decisively beat naive reporting and the classical Rogan–Gladen correction (Rogan & Gladen, 1978) (which pays a $1/J^2$ variance penalty), and match the gold-only mean on balanced low-bias gates. We thus recommend a PPI++ estimate reported with $J$ as the gate's *informativeness* coefficient—at $J=0$ the gated rate is uninformative and no estimator beats the gold-only mean—and take $K^\star=1$ as the conservative measurement cap (retrying only erodes $J$ in the pass-prone regime, and no larger cap improves recovery empirically; the theorem is the $1/J^2$ variance divergence as $J \to 0$).

- **Real agent loops, objective ground truth, three tasks (§5.1).** On 52 genuine revise-then-rejudge loops across GSM8K, MATH, and code generation (agent×judge×rubric), retrying inflates reported success up to 2.7× over objectively-true delivered success; 45–73% of retry gains on a fallible agent are judge slips, not fixes; the effect is natural (capable agents, strict rubrics) and *largest on code* (where the judge cannot run the candidate); even GPT-4.1 leaks; the correction recovers truth up to 11× better; and a closed form predicts the slip across all 52 configs ($r=0.95$). (The 2.7×/11× extremes are the terse stress agent; the capable-agent median slip is +0.16 and its worst group is code, true 0.74 to reported 0.98 (Table 3).)

- **Breadth: non-verifiable domains, a stopping-rule taxonomy, a transfer check (§5).** Beyond verifiable gold we *measure* recovery on public, human-labeled *non-verifiable* gates (response safety, summary quality), where PPI++ beats naive $\sim 4\times$ (the regime the method is built for); we show the bias generalizes across judge-aggregation rules (retry, majority, unanimous) with oppositely-signed bias; and we give a *label-free* detector (ROC-AUC 0.80) for when the correction can be trusted—across twelve public gates (17,280 rulings).

The takeaway for practitioners is a recipe, not just a warning: a loop-gated pass rate should be reported with its calibrated bias and reliability $J$ rather than raw; de-biasing helps decisively when that bias is large (and is within noise when it is small, where the diagnostic correctly says not to bother), the calibration needs only a handful of labeled items, and—counter to the intuition that more retries means a better system—in

the pass-prone regime where the loop bites (Prop. 1), more retries make the gate *less* informative, and (at leading order) do not help measurement (the recommended cap is $K^\star = 1$).

## 2 The loop-gated success rate is a biased estimator

**Setup.** An agent attempts $n$ items drawn i.i.d. Item $i$ has a true label $y_i \in \{\text{compliant}, \text{violation}\}$; let $\pi = \Pr[y_i = \text{compliant}]$ be the *true success rate* we wish to measure (estimated by the sample mean). A judge ruling on item $i$ returns PASS with item-specific probability $p_i$ (the judge's run-to-run instability—nonzero even at temperature zero, from non-associative floating-point reductions and server-side batching—estimated from repeated rulings). Under *retry-until-PASS* with cap $K$, item $i$ ships (ends in PASS) iff the judge passes it on at least one of up to $K$ independent rulings:

$$s_i(K) \;=\; \Pr[\text{ship within } K] \;=\; 1 - (1 - p_i)^K. \tag{1}$$

The reported success rate is $R(K) = \frac{1}{n}\sum_i s_i(K)$. Equation (1) models the $K$ rulings as independent draws with a fixed per-item rate $p_i$; we do not assume this, we check it. Across the public re-judgment gates (§5.2) the lag-1 same-verdict rate across reruns matches the value predicted under independence ($\mathbb{E}[\text{same} \mid p_i] = p_i^2 + (1 - p_i)^2$) to within 0.004 on every gate—no detectable serial correlation—and a substantial fraction of items are genuinely "flippy" ($0 < p_i < 1$) rather than deterministic. We estimate each $p_i$ from repeated rulings and propagate that finite-sample noise through a two-level bootstrap (§5), and we cap all empirical caps at the number of rulings so no quantity is extrapolated.

**Four estimands.** To keep the target unambiguous we name the quantities the loop touches. (i) *Single-attempt success* $\pi = \Pr[y_i = \text{compliant}]$: the quality of the agent's first attempt. (ii) *True delivered success* after cap $K$, $\pi_{\text{del}}(K) = R(K) \cdot \text{PPV}(K)$ with $\text{PPV}(K) = \Pr[\text{compliant} \mid \text{shipped}]$: the fraction of items that are *both* shipped and genuinely compliant (the quantity recovered in §5.1). (iii) The *reported judge-gated pass rate* $R(K)$: the observed, biased number. (iv) Our *corrected estimate* $\hat{\pi}$ of the target. In pure re-judgment of *fixed* work, $\pi_{\text{del}}(K)$ and $\pi$ coincide and $R(K)$ is a biased measurement of the single quantity $\pi$; in a genuine revise-then-rejudge loop later attempts can change the work, so $\pi_{\text{del}}(K)$ may exceed $\pi$ *to the extent revision genuinely fixes items*—then $K$ changes both the measurement procedure and the system measured. We are explicit about which estimand each result targets: $R(K)$-vs-$\pi$ bias on re-judgment gates (§5.2), and $R(K)$-vs-$\pi_{\text{del}}$ on real loops, decomposed into legitimate fixes vs. judge slips (§5.1).

**The gate is a classifier.** Group items by true label. Define the gate's class-conditional ship rates

$$\text{Se}(K) = \mathop{\mathbb{E}}_{y_i = \text{comp}} s_i(K), \qquad 1 - \text{Sp}(K) = \mathop{\mathbb{E}}_{y_i = \text{viol}} s_i(K), \tag{2}$$

i.e. $\text{Se}(K)$ is the probability a truly compliant item is (correctly) shipped and $1 - \text{Sp}(K)$ is the probability a true violation is (wrongly) shipped—the *slip* or leak rate. The cap-$K$ gate is then a binary classifier "ship = predict-compliant," and its expected apparent positive rate is the law of total probability:

$$\mathbb{E}[R(K)] \;=\; \pi\,\text{Se}(K) \;+\; (1 - \pi)\,\big(1 - \text{Sp}(K)\big). \tag{3}$$

Equation (3) is the crux: $R(K)$ equals $\pi$ only if the gate is perfect ($\text{Se} = \text{Sp} = 1$). For any imperfect judge it is a biased measurement whose bias, $R(K) - \pi = (1 - \pi)(1 - \text{Sp}(K)) - \pi(1 - \text{Se}(K))$, mixes two error channels—shipped violations and rejected compliant work—in proportions that move with the retry cap.

## 3 Retrying erodes the gate's discrimination

Both class-conditional ship rates rise with $K$ (Eq. 1 is increasing in $K$ for every $p_i$). What matters for measurement is their *difference*, the gate's Youden index

$$J(K) \;=\; \text{Se}(K) + \text{Sp}(K) - 1 \;=\; \mathop{\mathbb{E}}_{y_i = \text{comp}} s_i(K) \;-\; \mathop{\mathbb{E}}_{y_i = \text{viol}} s_i(K), \tag{4}$$

the standard scalar discrimination of a binary test (Youden, 1950).

**Proposition 1** (Discrimination is non-increasing in the retry cap)**.** *By definition* $J(K+1) - J(K) = \mathbb{E}_{comp}[\Delta s] - \mathbb{E}_{viol}[\Delta s]$, *with the per-attempt gain* $\Delta s(p) = s_i(K+1) - s_i(K) = p(1-p)^K$. *A* sufficient *condition for* $J(K+1) \leq J(K)$ *is that both class pass-rate distributions are supported on the decreasing branch* $p \geq 1/(K+1)$ *and the compliant class first-order stochastically dominates (FOSD) the violation class.*

*Proof.* $\Delta s(p) = p(1-p)^K$ is unimodal with peak at $p=1/(K+1)$, so on $p \geq 1/(K+1)$ it is monotone decreasing. FOSD of compliant over violation then reverses the expectation order for a decreasing function: $\mathbb{E}_{\text{comp}}[\Delta s] \leq \mathbb{E}_{\text{viol}}[\Delta s]$, i.e. $J(K+1) - J(K) \leq 0$. $\quad\square$

This condition is *sufficient*, not universal: it characterizes pass-prone gates—a property of where the two class-conditional pass-rate distributions sit, independent of the prevalence $\pi$ (which enters the bias of Eq. (3), not $J$)—rather than every gate. In that regime the gate's Sp collapses with retries while Se barely moves—each retry leaks more violations than it rescues compliant work—and $J(K)$ decreases monotonically; because both classes are then ever-passable, the limit is $J(\infty)=0$. Separately and unconditionally, $J(K)$ converges (by bounded convergence) to

$$J(\infty) = \Pr_{\text{comp}}[p_i > 0] - \Pr_{\text{viol}}[p_i > 0], \tag{5}$$

the gap between the fractions of each class the judge *ever* passes. A *nonzero* limit requires never-pass items (an atom at $p=0$), which lie *outside* Prop. 1's decreasing-branch hypothesis: there the per-attempt gain $\Delta s(p) = p(1-p)^K$ places mass on its increasing branch, the FOSD argument no longer applies, and $J$ can in fact *rise* with $K$—approaching $J(\infty)$ from below, a ceiling rather than a floor—and $J(\infty)$ can even be negative (an anti-discriminative gate). The two statements thus have disjoint hypotheses: monotone erosion holds in the pass-prone regime, where both classes are ever-passable and the only limit is $J(\infty)=0$, whereas a nonzero limit is an out-of-regime, possibly non-monotone phenomenon. Where it holds, erosion is the loop-level face of optional stopping—the stopping rule consumes the instrument's discriminating power—and we do not assert it universally. Measuring the exact controlling quantity $J(K+1)-J(K) = \mathbb{E}_{\text{comp}}[\Delta s]-\mathbb{E}_{\text{viol}}[\Delta s]$ on all twelve public gates—which are balanced ($\pi=0.5$)—it stays near-zero ($|\Delta J| < 0.02$); these balanced gates fall *outside* Prop. 1's hypothesis, so this near-zero value is *measured*, not predicted by the proposition (a sufficient condition is silent where its hypothesis fails). The synthetic gate of §4 instead *sweeps judge quality at a fixed* $K=1$ ($J : 0.80 \to 0.01$): it evidences the $J \to 0$ *non-identifiability* of §4, not erosion *with the retry cap*. We therefore present erosion-with-$K$ as a *gate-dependent* regularity, established by Prop. 1 and biting in the pass-prone regime where the loop is most consequential—not a universal law; on the real balanced gates it currently rests on Prop. 1 rather than on a measured $K$-sweep.

**Beyond retry: a taxonomy of stopping-rule bias.** Retry-until-PASS is one judge-aggregation rule among several, and the bias is a property of the family. Re-aggregating the same six rulings on each public gate, the signed bias is ordered by how pass-prone the rule is: any-pass/retry is the most inflationary, majority-vote intermediate, and unanimous the most conservative—so on a fixed judge and system the reported success rate is a free parameter of the aggregation choice, and two rules can give *oppositely-signed* bias on the same gate (e.g., GSM8K/GPT-4o-mini under a lenient rubric: any-pass +0.017, unanimous −0.033). Crucially, retry is *not* a fixed parallel-OR test: the cap is a *data-dependent stopping time*, so Se is inflated by the stopping rule itself and the classical fixed-Se/Sp screening algebra under-predicts the leak. Recovery is rule-agnostic—each rule induces its own Se, Sp—so the same $J$ diagnostic and PPI++/Rogan–Gladen correction apply throughout.

## 4 Recovering true success, and when it is possible

**The correction.** Equation (3) is linear in $\pi$ and inverts directly. Given the gate's class-conditional rates, the *Rogan–Gladen* estimator (Rogan & Gladen, 1978)—the classical correction of an observed prevalence for an imperfect diagnostic test—recovers true success:

$$\hat{\pi}(K) = \frac{R(K) - (1 - \text{Sp}(K))}{\text{Se}(K) + \text{Sp}(K) - 1} = \frac{R(K) - (1 - \text{Sp}(K))}{J(K)}. \tag{6}$$

| estimator | MAE @ 50% cal | @ 20% | @ 10% |
|---|---|---|---|
| naive (report gated rate) | 0.149 | 0.148 | 0.147 |
| Rogan–Gladen | 0.149 | 0.194 | 0.241 |
| Bayesian $(\mathrm{Se}, \mathrm{Sp}$ posterior) | 0.142 | 0.146 | 0.139 |
| gold-only mean (labeled subset) | 0.052 | 0.063 | 0.084 |
| PPI | 0.049 | 0.066 | 0.091 |
| PPI++ | 0.050 | 0.061 | 0.081 |

Table 1: Estimator bake-off: out-of-sample recovery MAE vs. true $\pi$ across the twelve re-judgment gates, by gold-calibration fraction (300 splits). The naive report and the higher-variance corrections (Rogan–Gladen, Bayesian) are decisively beaten by the label-efficient PPI estimators; Rogan–Gladen additionally pays the $1/J^2$ penalty (degrades to 0.241). Within the label-efficient block the PPI family and the gold-only mean lie within bootstrap noise of one another on these balanced gates—none uniformly best (§5.2)—so we recommend PPI++ because it never underperforms the gold-only mean (to which it reduces when the judge is uninformative) while retaining PPI's efficiency when the judge is informative; the gold-only mean and Rogan–Gladen also admit valid intervals (exact binomial and Reiczigel et al. (2010), respectively), and we report interval widths as the empirical split spread (App. A). We report $J$ as the identifiability diagnostic that says whether the gate adds information over gold at all.

In practice $\mathrm{Se}(K)$ and $\mathrm{Sp}(K)$ are unknown but estimable from a small *calibration* set of gold-labeled items run through the same gate; $\hat{\pi}$ is then applied to the unlabeled bulk. This is a measurement-design analogue of recent externally-valid judge estimators (Guerdan et al., 2025): a few labels fix the instrument's class-conditional behavior, and the rest of the data is corrected, not trusted raw.

**Which estimator, and the true role of $J$.** Rogan–Gladen is the most *interpretable* corrector—it exposes $\mathrm{Se}, \mathrm{Sp}$ and hence $J$—but it is not the most accurate, and saying so sharpens the contribution. Benchmarked out of sample against the prevalence-under-noise alternatives—prediction-powered inference (PPI/PPI++) (Angelopoulos et al., 2023a) and a Bayesian Se/Sp-uncertainty estimator—across all twelve gates the **label-efficient estimators (PPI/PPI++) decisively beat naive reporting and Rogan–Gladen**: mean recovery MAE 0.050 (PPI 0.049) vs. Rogan–Gladen 0.149 and naive 0.149 at a 50% calibration split, and the gap widens as gold becomes scarce (Rogan–Gladen 0.194/0.241 at 20%/10% calibration while PPI++ holds 0.061/0.081); on these balanced low-bias gates PPI, PPI++ and the gold-only mean lie within bootstrap noise of one another (none uniformly best, §5.2). The reason is structural: Rogan–Gladen's variance scales as $1/J^2$ (its empirical spread tracks $1/J$ across gates, $r=0.93$, and explodes at low $J$), whereas PPI++'s variance is governed by the judge–gold correlation and tracks $1/J$ far more weakly ($r=0.66$), so it *escapes* the $1/J^2$ penalty. This resolves an apparent tension between estimator and theory: $J$ is not the variance of any one estimator but the gate's *informativeness* coefficient—at $J=0$ the gated rate carries no information about $\pi$, so every estimator falls back to the gold-only mean and the gate's *efficiency* gain vanishes (gold still identifies $\pi$; it is the gate that becomes useless). We therefore recommend reporting **a PPI++ point estimate alongside $J$ as the reliability diagnostic**, with Rogan–Gladen as the bridge that makes $J$ explicit; $K^\star=1$ is the safe measurement cap for the whole pipeline: in the Prop. 1 regime retrying lowers $J$, and it never *improves* recovery for either estimator. We confirm this empirically (App. A): no larger cap improves out-of-sample recovery MAE for either estimator (PPI++ flat at 0.046; Rogan–Gladen 0.119 → 0.120 across $K=1$–6), so $K=1$ is the conservative choice for the recommended estimator too.

**Identifiability and the cost of retrying.** With the calibrated $\mathrm{Se}, \mathrm{Sp}$ treated as known, the delta method gives $\mathrm{Var}(\hat{\pi}(K)) \approx \mathrm{Var}(R(K))/J(K)^2$, and since $s_i \in [0,1]$,

$$\mathrm{Var}\big(\hat{\pi}(K)\big) \;\lesssim\; \frac{R(K)\big(1 - R(K)\big)}{n\,J(K)^2}. \tag{7}$$

Estimating $\mathrm{Se}, \mathrm{Sp}$ from a finite calibration set adds further terms that also scale with $1/J^2$; we therefore do not rely on Eq. (7) for our intervals but report the empirical spread of the out-of-sample estimator

(§5), which includes calibration noise. The qualitative content is what matters: the recovered estimate's precision scales with $1/J^2$. For any $J > 0$, $\pi$ is point-identified by Eq. (6); as $J \to 0$ the inversion becomes ill-conditioned and $\pi$ is non-identifiable exactly at $J=0$, where two agents of different quality give the same gated rate. A synthetic gate makes this concrete (compliant pass-rate 0.90, violation pass-rate $q \to 0.90$): $J$ falls $0.80 \to 0.01$, the gated rates of two agents differing by 0.20 in true success collapse from a 0.16 gap to 0.002, and $1/J$ explodes to $100\times$. Real gates sit above this floor (§5.2: $J=0.16$–$0.45$), so we see inflated variance, not collapse—but the floor is what a practitioner must monitor.

Because $J(K)$ is non-increasing in the pass-prone regime (Prop. 1) and retrying cannot raise it there, and because empirically no larger cap improves recovery for either estimator (App. A), **we recommend $K^\star = 1$ as the conservative measurement cap**. This is an operational default, not a strict variance minimum: the variance $V(K)/J(K)^2$ need not be monotone in $K$ (the numerator can shrink as the gate saturates); what is a theorem is the $1/J^2$ *divergence* as $J \to 0$. This yields a two-part recipe. *Prevention:* to *measure* success, run the gate once and report the corrected estimate—in the pass-prone regime retrying only erodes $J$. *Cure:* when a system is already locked into a high-retry loop (the rate cannot be re-collected at $K=1$), or when the gate is biased even at $K=1$, the correction recovers what the raw rate cannot. The distinction from agent *utility* is deliberate: a retry policy may well help the agent succeed; our claim is only about *measuring* that success.

**A label-free transfer check.** The correction is only as good as the calibration set's transfer to the evaluation set, and this can fail—calibrating on a 0.96-solve agent and recovering a 0.30-solve agent gives MAE 0.31, worse than naive. We give a deploy-time detector that needs *no* evaluation gold: the observed gated-rate shift $|R_{\text{cal}} - R_{\text{eval}}|$. Across 76 cross-agent transfers it predicts whether the correction will beat naive with ROC-AUC 0.80; trusting the correction only when the shift is below 0.05 gives precision 0.94, and one falls back to the raw rate (or re-calibrates) above it. We report this threshold as an *illustrative* operating point computed in-sample over the 76 transfers, not an out-of-sample-validated guarantee; the takeaway is that a cheap, label-free signal carries real information about transfer failure. Calibration transfer is thus a *monitorable operating condition*, not a silent assumption. The detector is *necessary, not sufficient*: similar marginal gated rates do not *guarantee* stable Se, Sp (the confusion structure can shift while $R$ is unchanged), so a passing check licenses de-biasing only together with the $J$/bias diagnostic, never on its own.

## 5 Experiments

We study the loop in three settings: **real revise-then-rejudge agent loops** with objective ground truth (§5.1, the most direct test), a **cross-judge study** over twelve public judges (§5.2), and a **non-verifiable, human-labeled** domain (§5.3). Throughout, "naive" means reporting the gated rate as the success rate, and the out-of-sample protocol estimates the calibration quantities on a random gold-labeled half and recovers on the held-out half (300 splits).

### 5.1 Real revise-then-rejudge agent loops

The closed-form analysis above models retries as re-judgments of fixed work. We now test the genuine case—an agent that *revises* between attempts—with *objective* ground truth. On GSM8K, an agent solves a problem; an LLM judge gates PASS/FAIL; on FAIL the agent regenerates with rejection feedback and is re-judged, to a cap $K=5$; the loop ships the first PASS. True correctness is exact match to the gold integer (no human gold, no $\kappa$). We cross three agent-capability levels (a terse no-scratchpad agent and careful GPT-4o-mini and GPT-4o; objective solve rates $\approx 0.34/0.92/0.96$), four judges (GPT-4o-mini, GPT-4o, GPT-4.1, Gemini-2.5-Flash), and two rubrics (strict; lenient "approve unless clearly wrong")—24 GSM8K loop configurations, $n=250$, $\sim$30k agent+judge calls; with the MATH (16) and code (12) sweeps below this totals 52 loop configurations (44 excluding the deliberate terse stress agent). Because we log every attempt's verdict *and* objective correctness, we can replay any cap, decompose retry gains, and recover true success.

**Retrying inflates reported success far above truth.** A fallible agent (terse; true success $\approx 0.29$) gated by a GPT-4o judge reports success climbing from 0.62 to 0.78 as $K$ goes $1 \to 5$, while objectively-true

| agent (true solve rate) | judge (strict rubric) — slip $R-\pi$ at $K{=}5$ | | | |
|---|---|---|---|---|
| | GPT-4o-mini | GPT-4o | GPT-4.1 | Gemini-Flash |
| terse ($\approx 0.34$) | $+0.15$ | $+0.49$ | $+0.46$ | $+0.38$ |
| careful GPT-4o-mini ($\approx 0.92$) | $+0.06$ | $+0.07$ | $+0.07$ | $+0.07$ |
| careful GPT-4o ($\approx 0.96$) | $+0.03$ | $+0.04$ | $+0.04$ | $+0.02$ |

Table 2: Real agent loops, slip (reported $-$ objectively-true delivered success) at $K{=}5$. Slip scales with agent failure $\times$ judge leak: large for a fallible agent, small for a strong one. Judge discrimination $J$ against objective truth spans 0.007 (GPT-4o-mini/lenient—a coin flip) to 0.65; even the strongest judge (GPT-4.1, $J{\approx}0.25$–0.37) has nonzero leak, so no judge eliminates the slip.

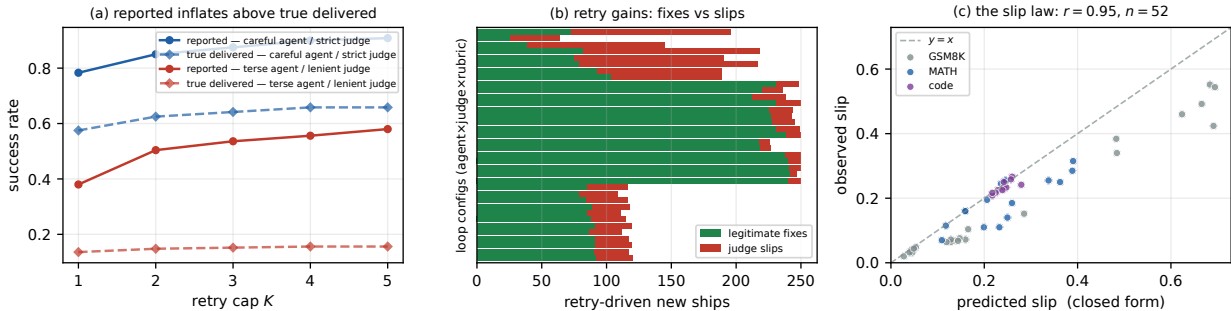

Figure 1: Real agent loops. **(a)** Reported success inflates above objectively-true delivered success as the loop retries (pathological: terse agent / lenient judge; benign: careful agent / strict judge). **(b)** Retry-driven new ships split into legitimate fixes vs. judge slips—on the fallible agent, 45–73% of retry gains are slips. **(c)** The slip obeys a predictive *law*, not a corner: a parameter-free forward model from per-attempt $(a, s, \ell)$ tracks observed slip across all 52 configs (pooled $r{=}0.95$) and within each task (mean $r{=}0.86$); regression slope 0.74 ($< 1$; the gap reflects revision net of selection), confirming it is a real prediction, not the slip=reported$\times(1{-}$PPV) identity.

delivered success barely moves ($0.26{\to}0.29$): a slip of $+\mathbf{0.49}$. Decomposing the retry-driven new ships, 63% **are judge slips**—the judge waved through a still-wrong answer—not legitimate fixes. The agent is not improving; retrying manufactures false successes by giving an imperfect judge repeated chances to pass a wrong answer. Manual inspection confirms the mechanism: the terse agent emits a bare wrong number (e.g. `Answer:  65000` when the truth is 70000) and the judge, with no reasoning to check, rubber-stamps it—so *agent capability modulates the judge's leak*, not merely its own correctness (the careful agent, which shows work, draws a markedly lower leak). We stress that these loop quantities are computed *directly* from each attempt's logged verdict and objective correctness; they do not rely on the closed-form draw model of §2 (which we invoke only for the fixed-work re-judgment gates). This terse/lenient config is a deliberate *stress case*; the effect is not an artifact of an artificially weak agent: a capable agent (GPT-4o-mini, $\approx 0.92$ solve rate) under the same judges still slips a real $+0.06$ to $+0.10$, and the strong GPT-4o agent $+0.02$ to $+0.05$ (Table 2)—small but nonzero wherever the judge leaks, which is everywhere.

**Recovery is decisive here.** The loop ships only PASS, so true delivered success is reported $\times$ PPV, with PPV = Pr[correct | shipped] estimable from a small gold-labeled calibration set. Out of sample, the corrected estimate's error is *bounded*—MAE 0.02–0.05 across all 24 configs—while the naive report's error tracks the slip, up to 0.55. On the headline config the correction is $\mathbf{11\times}$ more accurate (MAE 0.044 vs. 0.494). Unlike the balanced re-judgment gates of §5.2 (where the per-gate win was within noise), here the slip is large and the correction's advantage is decisive and significant. The benign corner—strong agent, accurate judge—shows small slip and little need for correction, exactly as the reliability diagnostic prescribes: report $J$ and the calibrated bias, and correct when they are large. The bootstrap intervals are well-calibrated: on a synthetic gate with known $\pi$, empirical 95%-CI coverage stays in $[0.92, 0.97]$ across $J \in [0.1, 0.8]$, and the

| task | rubric | $n$ | true $\pi$ | reported | mean slip | slip range | MAE corr/naive |
|------|--------|-----|------------|----------|-----------|------------|----------------|
| GSM8K | strict | 8 | 0.92 | 0.97 | +0.05 | [+0.02, +0.07] | 0.02/0.05 |
| GSM8K | lenient | 8 | 0.92 | 0.98 | +0.06 | [+0.03, +0.10] | 0.02/0.06 |
| MATH | strict | 8 | 0.71 | 0.86 | +0.14 | [+0.07, +0.26] | 0.04/0.14 |
| MATH | lenient | 8 | 0.71 | 0.95 | +0.24 | [+0.16, +0.31] | 0.05/0.24 |
| code | strict | 6 | 0.72 | 0.95 | +0.23 | [+0.21, +0.26] | 0.06/0.23 |
| code | lenient | 6 | 0.74 | 0.98 | +0.24 | [+0.22, +0.27] | 0.06/0.24 |

Table 3: The slip across capable-agent loops ($K{=}5$), grouped by task and rubric—the full distribution behind the headline. Slip is task-ordered (small on GSM8K, large on code where the judge cannot execute) and present even under *strict* judging. In every cell the calibrated correction (corr) beats the naive report, cutting MAE $\sim 4\times$ overall (mean 0.04 vs 0.16). The terse stress case (Table 2) is excluded here.

reliability floor manifests as interval *width* ($\propto 1/J$: 0.20 at $J{=}0.8$ growing to 0.92 at $J{=}0.1$), not as broken coverage.

**The inflation is a distribution, not a corner: MATH and code.**   The headline numbers above are deliberate stress cases, so the fair question is what the slip looks like *typically*. To answer it we replicate on competition MATH (integer-answer subset) and code generation (MBPP, with *execution* ground truth—the canonical agentic setting) using only *capable* agents (natural solve 0.5–0.96) and no terse trick, and report the full per-task distribution (Table 3). Across all 44 capable-agent configurations the median slip is +0.16 and the mean +0.15 (range +0.02 to +0.32)—so the effect is not a single adversarial corner but a systematic, task-ordered bias: small on easy GSM8K (+0.05), substantial on MATH even under the *strict* rubric (+0.14), and largest on *code* (+0.23, $J$ as low as 0.07), where a judge that cannot run the candidate rubber-stamps plausible-looking programs. Critically, the correction is decisive across the *whole* distribution, not just the corners: it recovers true delivered success to MAE 0.04 versus naive 0.16 on average—a $\sim 4\times$ reduction that holds in every task–rubric cell of Table 3.

**A closed-form law predicts the slip.**   What makes the bias a measurement quantity rather than a list of anecdotes is that it obeys a *predictive law*—and we are careful to separate what is definitional from what is empirical. The accounting identity slip $=$ reported $\times (1 - \mathrm{PPV})$ holds by construction for any ship-on-PASS loop and is *not* the claim. The empirical claim is that a parameter-free *forward* model predicts the slip from the loop's per-attempt statistics: modeling each attempt as independently correct with rate $a$, judge sensitivity $s$, and leak $\ell$ gives reported$(K) = 1 - (1-\rho)^K$ and slip$(K) = $ reported$\times (1-a)\ell/\rho$ $(\rho{=}as+(1-a)\ell)$. Each config's $(a, s, \ell)$ are estimated only from *that config's own* attempt logs—no parameters are shared or fit across configs. Because a config's parameters are otherwise fit and evaluated on the same logs, we also run a stricter *item-level* held-out test: fitting $(a, s, \ell)$ on a random half of each config's items and predicting the slip on the *disjoint* held-out half (200 splits) gives held-out slip correlation $r{=}0.95$ and held-out slip-MAE 0.07, so the law generalizes across items rather than merely refitting its own logs. The prediction tracks the slip measured against *objective* truth not only across all 52 configs (pooled $r{=}0.95$, bootstrap 95% CI [0.92, 0.97]) but *within* each task (mean within-task $r{=}0.86$: GSM8K 0.99, MATH 0.82, code 0.78; Fig. 1c), so the fit is not an artifact of between-task spread, and the predictive slip-MAE is 0.05. Decisively, the predicted-vs-observed regression slope is 0.74 with bootstrap 95% CI [0.67, 0.80]—excluding 1 ($p < 0.001$), which a definitional identity would force—so the law is a genuine forward prediction, not the slip$=$reported $\times (1 - \mathrm{PPV})$ identity restated. The i.i.d.-attempt model systematically *over*-predicts slip by this margin; that residual is the *net* effect of the agent's revision and of retry-induced selection (later attempts exist only for previously-failed items), so we read it as an upper bound on the revision benefit rather than the revision benefit exactly. The law thus turns the path-dependence of revised attempts into a measured quantity rather than an unmodeled assumption, and confirms retries mostly manufacture slips rather than fixes. A regression-dilution check rules out the innocuous alternative that the $< 1$ slope is mere predictor noise: the predictor reliability is 0.977 and the errors-in-variables (Deming) slope is 0.765 [0.70, 0.84], still excluding 1—this rules out attenuation from predictor noise, though not selection—and the slope stays below 1 *within* task (GSM8K 0.76, MATH 0.79; code $\approx 1$, i.e. revision cannot help where the judge cannot execute), the within-task code cell being the

| gate (judge, lenient grading) | Se(1) | leak $1-$Sp | true $\pi$ | bias $R-\pi$ |
|---|---|---|---|---|
| GSM8K / Gemini-Flash | 0.969 | 0.733 | 0.50 | +0.351 |
| TruthfulQA / GPT-4o-mini | 0.821 | 0.596 | 0.50 | +0.208 |
| HaluEval / Gemini-Flash | 0.844 | 0.431 | 0.50 | +0.137 |
| TruthfulQA / Gemini-Flash | 0.824 | 0.404 | 0.50 | +0.114 |
| HaluEval / GPT-4o-mini | 0.690 | 0.394 | 0.50 | +0.042 |

Table 4: The pass-prone inflation regime on public data. Lenient judges leak violations (leak $\approx$ 0.4–0.7) and inflate reported success above true success—by up to +0.35. Across all twelve gates the calibrated bias spans $-0.35$ to $+0.35$ (both directions present publicly).

cleanest evidence for the revision reading. Per task×rubric cell, the recovery gain (naive−corrected MAE) is positive in all six cells with bootstrap CIs excluding zero (Holm-corrected); the within-noise outcome is confined to the balanced re-judgment gates, where a power analysis shows the evaluation ($n{=}240$) can only resolve gaps above $\approx 0.009$.

**The motivating regime, measured: a powered, pre-registered non-verifiable loop.** The setting the method is built for is a non-verifiable gate *inside* a real loop—a safety/compliance judge that cannot mechanically check the work, where a slip is a shipped *unsafe* response. We measure it directly and at scale on BeaverTails ($n{=}400$ prompts): an agent revises under a weak gpt-4o-mini safety gate to cap $K{=}5$, with ground truth a *pre-registered* 3-judge strong panel (GPT-4.1, Claude, Gemini; majority-safe, the rule fixed in advance). The phenomenon is clear and statistically decisive. A *lenient* gate ships every response ($J{=}0$) yet 6.8% are *panel-labeled* unsafe (slip +0.068, Wilson 95% CI [0.047, 0.096], excluding zero); a *strict* gate ($J{=}0.35$) shows the slip *grow with retries*—+0.037 at $K{=}1$ to +0.042 at $K{=}5$ (reported $0.91 \to 0.98$, panel-safe $0.88 \to 0.94$; 4% of retry-driven ships are judge slips), the optional-stopping inflation in its motivating form. In both, the calibrated PPV correction is *decisively* bounded (MAE 0.017–0.019) while naive tracks the slip (0.04–0.07): a 2.4–3.5× recovery, now in the non-verifiable, looped regime rather than only on verifiable tasks. We anchor the panel gold to humans: on a 120-response subset a human labeler agrees with the panel at raw 0.90 (Cohen's $\kappa{=}0.45$, depressed by the $\sim 7\%$ unsafe base rate; PABAK 0.80), with a human-gold slip (+0.042) of the same sign and order. The weak in-loop judge agrees with humans far less ($\kappa{=}0.15$)—it rubber-stamps the safe majority while missing the unsafe minority, exactly the leak we study. Honest scope: ground truth here is a strong-model panel human-validated on a subset, not human adjudication at full scale—a remaining annotation cost, not a missing result.

## 5.2 A cross-judge measurement study

The framework also applies to gates that re-judge fixed work, where the closed-form $\text{Se}(K), \text{Sp}(K)$ hold directly. To test the account across many judges on fully auditable data, we replicate the full pipeline across **twelve public gates** (dataset × judge × grading-style) over three datasets—TruthfulQA (Lin et al., 2022), HaluEval-QA (Li et al., 2023), GSM8K (Cobbe et al., 2021)—two judges (GPT-4o-mini, Gemini-2.5-Flash), and two grading styles (a strict correctness rubric and a *lenient* "approve unless clearly wrong" rubric), each ruling six times at temperature zero (17,280 rulings; same correct-vs-incorrect-answer construction). Independence held on every gate (lag-1 same-verdict within 0.004 of the i.i.d. prediction); $J(1)$ ranges 0.16 (GPT-4o-mini grading math) to 0.45.

**Both inflation and deflation appear publicly.** Under a strict rubric public judges are fail-prone (retries move *toward* truth); under a lenient rubric they are pass-prone and inflate reported success above truth (Table 4). This places the phenomenon on fully reproducible data, in both bias directions.

**Naive error is bias; correction trades it for variance.** Two facts across the twelve gates. First, near-definitionally, the naive report's error equals the gate's calibrated bias (corr = 0.998; both are $|R(1) - \pi|$ on gold)—we state it not as a discovery but to register that this bias is *large and real*, up to 0.35, and present on public gates. Second, and genuinely empirical: the de-biased estimate's error is decoupled from bias

| domain | judge / rubric | $\pi$ | naive bias | naive | RG | PPI++ |
|---|---|---|---|---|---|---|
| safety | gpt4o-mini / strict | 0.50 | $-0.04$ | 0.038 | 0.052 | 0.041 |
| safety | gpt4o-mini / lenient | 0.50 | $+0.10$ | 0.097 | 0.054 | **0.038** |
| safety | gemini / strict | 0.50 | $-0.21$ | 0.210 | 0.102 | **0.044** |
| safety | gemini / lenient | 0.50 | $+0.09$ | 0.095 | 0.057 | **0.039** |
| quality | gpt4o-mini / strict | 0.62 | $-0.54$ | 0.538 | 0.204 | **0.050** |
| quality | gpt4o-mini / lenient | 0.62 | $-0.22$ | 0.226 | 0.090 | **0.049** |
| quality | gemini / strict | 0.62 | $-0.10$ | 0.096 | 0.060 | **0.045** |
| quality | gemini / lenient | 0.62 | $+0.01$ | 0.020 | 0.055 | 0.040 |

Table 5: Recovery on *non-verifiable*, human-labeled gates (BeaverTails safety; SummEval quality), out of sample vs. human gold. Bias runs both ways (strict judges fail-prone, lenient pass-prone); PPI++ recovers the true rate to MAE $\approx 0.04$ throughout—$\sim 4\times$ better than naive on average and $>10\times$ on the most biased gate. The two gates where naive ties are exactly those with $|\text{bias}| \leq 0.04$, where the bias/$J$ diagnostic says not to de-bias.

(corr $= 0.17$) and instead tracks inverse reliability, corr(MAE, $1/J$) $= 0.93$ (bootstrap 95% CI $[0.88, 1.00]$; 0.99 with the GSM8K gates removed, so not an outlier artifact). So the correction does exactly one thing: it removes the bias and pays $1/J^2$ variance. Whether that is a net win on a *single* gate depends on whether $|\text{bias}|$ exceeds the recovery noise, and at our evaluation size ($n$=240) the per-gate paired MAE differences are within noise (bootstrap CIs include zero). We therefore do not claim a per-gate win; we claim the operative quantities. The bias is large, real, and exactly measured (up to 0.35), and $J$ governs the cost of removing it. *Report both:* trust the raw rate when the calibrated bias is small; de-bias when it is large and $J$ is not too low; and when $J$ is very low (GSM8K/GPT-4o-mini, $J$=0.16), the irreducible recovery error is itself the verdict that the judge is too unreliable to gate on. The reliability coefficient and the calibrated bias, not a single point estimate, are the deliverable. (Recovery also holds at skewed prevalence: subsampling public gates to $\pi$=0.2 and 0.8 still recovers the true rate, e.g. $\hat{\pi}$=0.80 at $\pi$=0.8.)

### 5.3 Recovery in a non-verifiable domain (safety and summary quality)

All evidence above uses *verifiable* gold (exact match, execution), yet the motivating regime—a compliance gate—is precisely one where the judge *cannot* mechanically check the work. We therefore test recovery directly on two public, *human*-labeled, non-verifiable gates: response **safety** (BeaverTails (Ji et al., 2023), $n$=300, human `is_safe` gold) and **summary quality** (SummEval (Fabbri et al., 2021), $n\approx$240, expert consistency scores, binarized), each judged six times at temperature zero under strict and lenient rubrics by two judges. These are re-judgment gates over fixed responses; combining non-verifiable judging *with* a revise-then-rejudge loop is measured in §5.1, anchored on human gold ($n$=120 shipped responses) with a strong-model panel used for scale (Table 6). The bias here runs in *both* directions: strict safety/quality judges are *fail*-prone (a strict summary judge under-passes good summaries, naive bias up to $-0.54$), lenient judges pass-prone—and naive reporting inherits whichever it is. Out of sample (50% human-gold calibration, 300 splits), **PPI++ recovers the true rate to mean MAE** 0.043 **versus naive** 0.165 across the eight gates ($\sim 4\times$ better; $>10\times$ on the most biased, where naive MAE 0.54 falls to 0.05), beating naive on 6/8 gates. The two exceptions are exactly the gates where the bias is already small ($|R - \pi| \leq 0.04$)—the regime where the bias/$J$ diagnostic correctly says *not* to de-bias. This moves the central claim from a structural argument to a *measured* result in the non-verifiable, human-gold regime the method is built for. As a third-provider robustness check, an Anthropic Claude safety judge (a third model family beyond the OpenAI and Google judges) is *well-calibrated*: $J \approx 0.58$ and $|\text{bias}| \leq 0.04$ on BeaverTails (strict/lenient), so naive ties the correction (PPI++ MAE 0.042). This is exactly the low-bias regime where the $J$/bias diagnostic says *not* to de-bias—and across all three judge families PPI++ is never worse than naive, confirming the selectivity of the recipe rather than a blanket correction.

# 6 Related work

**Measurement science and judge reliability.** A growing line treats benchmark scores as measurements with explicit validity and reliability (Jacobs & Wallach, 2021; Salaudeen et al., 2025; Polo et al., 2024), and documents that LLM judges flip across reruns and design choices (Feuer et al., 2025; Zheng et al., 2023); Guerdan et al. (2025) correct judge estimates for rating indeterminacy in the *isolated* judging setting; concurrent single-shot debiasing of imperfect judges includes Lee et al. (2025) (sensitivity/specificity bias with calibration-aware intervals), Chen et al. (2026) (a semiparametric-efficient estimator), and Collot et al. (2025) (Youden's $J$ to *select* a judge)—each scoring a judge that rules *once* per item. We move one level out, to the judge *embedded in a stopping rule*, where the bias is generated by the loop and $J$ becomes a cap-dependent $J(K)$ that retrying erodes—supplying the reliability coefficient and the correction it implies.

**Optional stopping and verifier-guided loops.** Retry-until-PASS is optional stopping against a noisy classifier, sharing its pathology with repeated significance testing (Armitage et al., 1969), adaptive holdout reuse (Dwork et al., 2015), post-selection inference (Berk et al., 2013; Gelman & Loken, 2013), and—in ML evaluation—adaptive overfitting to a reused test signal. Verifier-guided methods (Reflexion, self-refine, best-of-$N$) gate attempts by a learned/LLM verifier (Shinn et al., 2023; Madaan et al., 2023); closest, Dorner et al. (2025) characterize how verifier imperfection bounds the *achievable accuracy* of resampling. We ask the inverse, *measurement* question: their object is a policy's attainable performance (an optimization target), ours the estimator's bias and identifiability (a measurement defect). They give no correction recovering true prevalence from a gated rate, nor identify $J$ as the quantity that governs when recovery is possible. To our knowledge no prior work isolates the *stopping rule* (rather than any single ruling) as the generator of the measurement bias, nor identifies the *cap-dependent $J(K)$* that retrying erodes as the coefficient governing recoverability in the loop.

**Quantification, label shift, and prediction-powered inference.** Recovering true success $\pi$ from a noisy gated rate is formally the *quantification* problem—estimating class prevalence in aggregate rather than per item (Forman, 2008; González et al., 2017)—and our class-conditional correction is exactly Adjusted Classify & Count / Rogan–Gladen, with $\mathrm{Var} \sim 1/J^2$ making Youden's $J$ the visible reliability of the count. The calibration-transfer condition the correction rests on (a stable confusion structure from calibration to deployment) is the *label-shift* assumption (Saerens et al., 2002; Lipton et al., 2018; Alexandari et al., 2020), and our label-free $|R_{\mathrm{cal}} - R_{\mathrm{eval}}|$ diagnostic is a quantification shift test in the spirit of BBSE. For the point estimate we adopt PPI++ (Angelopoulos et al., 2023a;b), which improves on Rogan–Gladen by escaping its $1/J^2$ variance (§4). Recent work debiases a *single* noisy judge (Lee et al., 2025; Chen et al., 2026), uses Youden's $J$ to *select* a judge (Collot et al., 2025), or proposes $J$ and $\Delta J$ as reliability diagnostics for single-shot corrected judge estimates (Fiedler, 2026)—all in the single-ruling regime; we instead study the *cap-dependent $J(K)$* that a stopping rule erodes. Closest to our taxonomy, Ullah & Serwadda (2026) derive the operating characteristics of the *unanimous* jury—one cell of our rule taxonomy, which we generalize and correct; for deployment monitoring, Zhang et al. (2026a) give anytime-valid risk bounds under shift, but consume labels, whereas our transfer gate is label-free. Concurrently, Zhang et al. (2026b) report a Rogan–Gladen undercount in a single deployed LLM-judge gate (degenerating at zero apparent defects, the $J=0$ boundary of Eq. (6)); we instead isolate the stopping rule as the bias generator and give the cap-$K$, $J(K)$-diagnosed recovery recipe.

# 7 Limitations

We state each central claim with the regime where it is *measured* versus argued (Table 6).

The recovery rests on an identifying assumption: the gate's class-conditional rates (Se, Sp, or PPV in a loop) estimated on the calibration set must *transfer* to the evaluation set. We tested this directly—calibrating on one agent and recovering a very different one (a $0.96 \rightarrow 0.30$ solve-rate shift) degrades loop recovery to MAE 0.31, worse than naive—so a practitioner must calibrate on data representative of the system measured; within-distribution it is accurate. Our genuine revise-then-rejudge evidence with objective ground truth covers GSM8K, MATH, and code—all *verifiable* domains; recovery in the non-verifiable regime is validated on

| claim | strongest evidence | scope / caveat |
|---|---|---|
| bias exists, is $K$-dependent | 52 loops + 12 public gates | holds throughout |
| recovery: PPI++ $\gg$ naive | verifiable *and* non-verifiable human-gold gates | needs calibration transfer (gated, §4) |
| $J$-erosion with retries | Prop. 1 (pass-prone regime) | pass-prone gates; $\approx$flat ($|\Delta J|{<}0.02$) on balanced public gates |
| slip law; revision+selection residual | 52 loops, held-out $r{=}0.95$; Deming slope $0.765\,[0.70, 0.84]$ | verifiable-domain attempt logs |
| non-identifiability ($J{\to}0$) | synthetic + low-$J$ real gates ($J{\geq}0.16$) | full $J{=}0$ only synthetic |

Table 6: Claim $\times$ evidence scope: each central claim, its strongest support, and the regime where it is established rather than argued.

human-labeled re-judgment gates (§5.3) and the powered in-loop safety study (§5.1), but we do not validate it on open-ended multi-step trajectories, where the extension rests on the structural argument rather than a measured result. On the balanced re-judgment gates the per-gate net MAE win is within bootstrap noise at $n{=}240$ (decisive only on the high-slip real loops), so we report the bias and $J$ as the deliverables. The $J{\to}0$ non-identifiability regime is established by a synthetic low-$J$ gate rather than measured on a real one (our public gates are balanced with $J \geq 0.16$); erosion *with the retry cap* rests on Prop. 1 in the pass-prone regime and is $\approx$flat on the balanced public gates. Finally, the public re-judgment gates use a controlled lenient rubric and single-shot judging; we model such retries as re-judgments of fixed work, whereas a retry that genuinely *repairs* a violation is a legitimate pass, not a slip—our target is the measurement of success, not the agent's policy.

## 8   Conclusion

A loop-gated pass rate is a measurement taken with an unstable instrument and a stopping rule that consumes the instrument's discrimination. Read raw, it is biased, and it becomes non-identifiable exactly where the gate's discrimination vanishes ($J{=}0$)—a floor that retrying drives toward only in the pass-prone regime, since $J(\infty)$ need not otherwise be zero. The conceptual shift is to read it not as a number but as the output of an instrument with a known reliability: its bias is then a calibrated, removable quantity, and $J$ says in advance whether removing it is worth the variance. Two numbers—the calibrated bias and the reliability $J$—replace one untrustworthy point estimate. The guidance is crisp: measure at $K{=}1$, report $J$ and the calibrated bias, de-bias when that bias exceeds the recovery standard error ($\propto 1/J$), and treat a very low $J$ as itself the verdict that the gate is too unreliable to trust. We release the estimator and content-free data (per-item judge verdicts, gold labels, and configuration—no raw prompts or model responses, so exact re-generation of the underlying text requires re-running the judges) to make this the default.

## Broader Impact Statement

This work is a measurement-and-auditing method: it exposes that LLM-judge-gated success rates—including safety gates that decide which responses ship—are biased (upward in the pass-prone retry regimes that motivate this work) and, at the reliability floor $J{=}0$, non-identifiable, and it supplies a calibrated correction. The intended effect is to reduce harm, by making it harder for genuinely unsafe or non-compliant outputs to be reported as "passed." We see two risks to flag. First, our method is a *distrust-generating diagnostic, not an endorsement mechanism*: a calibrated estimate must not be read as legitimizing an LLM-judge safety gate. Both a low $J$ *and* a failed calibration-transfer check are evidence *not to rely on the gate*—not merely situations to correct—and we recommend reporting the reliability verdict alongside any corrected estimate rather than shipping on the estimate alone. Second, our non-verifiable evaluation uses potentially unsafe prompts (BeaverTails) and a strong-model panel as a scalable gold proxy human-anchored on a subset; the gold is not human adjudication at full scale, so reported absolute harm rates should be treated as panel-relative estimates, not certified ground truth. No new harmful capability or dataset is released; we release only the estimator and content-free (label/score-only) data.

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

# A    Finite-sample variance, the reliability floor, and $K^\star = 1$

The main-text remark after Eq. (7) propagates only the bulk-rate sampling error and treats the calibrated $\mathrm{Se}(K), \mathrm{Sp}(K)$ as known. Here we discharge that simplification: we give the full delta-method variance of the Rogan–Gladen estimator

$$\hat{\pi}(K) = \frac{R(K) - \big(1 - \mathrm{Sp}(K)\big)}{\mathrm{Se}(K) + \mathrm{Sp}(K) - 1} = \frac{R(K) - \big(1 - \mathrm{Sp}(K)\big)}{J(K)}$$

when $\mathrm{Se}(K)$ and $\mathrm{Sp}(K)$ are themselves *estimated* from a finite gold-labeled calibration set, and we show every variance term is $\Theta\big(1/J(K)^2\big)$. This is the finite-sample correction underlying the exact prevalence interval of Reiczigel et al. (2010). We then show the variance *diverges* as $\Theta(1/J(K)^2)$ at the reliability floor ($J \to 0$ at fixed non-degenerate numerator), so a gate driven to low $J$ is non-identifiable; we recommend $K^\star = 1$ as a conservative operational default (Remark 2), not as a proved variance minimum—the numerator $V(K)$ can shrink as the gate saturates, so the variance need not be monotone in the cap.

**Calibration design.**    Fix a cap $K$. The unlabeled bulk supplies $R \equiv R(K) = \frac{1}{n} \sum_i s_i(K)$ over $n$ items. A disjoint calibration set of $m$ gold-labeled items ($m_1$ truly-compliant, $m_0$ truly-violating) run through the *same* cap-$K$ gate yields

$$\widehat{\mathrm{Se}} = \frac{1}{m_1} \sum_{i:\, y_i=\mathrm{comp}} s_i(K), \qquad \widehat{\mathrm{Sp}} = 1 - \frac{1}{m_0} \sum_{i:\, y_i=\mathrm{viol}} s_i(K).$$

The three estimators are computed on disjoint item sets and are mutually independent, with binomial-type variances $\mathrm{Var}(\widehat{R}) = R(1-R)/n$, $\mathrm{Var}(\widehat{\mathrm{Se}}) = \mathrm{Se}(1-\mathrm{Se})/m_1$, $\mathrm{Var}(\widehat{\mathrm{Sp}}) = \mathrm{Sp}(1-\mathrm{Sp})/m_0$.

**Lemma 1** (Finite-sample delta-method variance)**.** *Write $J = J(K) = \mathrm{Se} + \mathrm{Sp} - 1$ and assume $J > 0$. With $\widehat{R}, \widehat{\mathrm{Se}}, \widehat{\mathrm{Sp}}$ independent as above, the first-order delta method gives*

$$\mathrm{Var}\big(\hat{\pi}(K)\big) \;\approx\; \frac{1}{J^2}\left[\frac{R(1-R)}{n} + \pi^2 \frac{\mathrm{Se}(1-\mathrm{Se})}{m_1} + (1-\pi)^2 \frac{\mathrm{Sp}(1-\mathrm{Sp})}{m_0}\right]. \tag{B.1}$$

*Every term carries the same $1/J^2$ factor; the two calibration terms are the finite-$m$ corrections omitted from Eq. (7).*

*Proof.* Treat $\hat{\pi} = g(\widehat{R}, \widehat{\mathrm{Se}}, \widehat{\mathrm{Sp}})$ with $g = (R-1+\mathrm{Sp})/(\mathrm{Se}+\mathrm{Sp}-1)$. With $J = \mathrm{Se}+\mathrm{Sp}-1$ and $R-(1-\mathrm{Sp}) = \pi J$ at the truth, $\partial g/\partial R = 1/J$, $\partial g/\partial \mathrm{Se} = -\pi/J$, and $\partial g/\partial \mathrm{Sp} = (1-\pi)/J$. By independence the delta method sums the squared derivatives times the binomial variances; factoring the common $1/J^2$ yields (B.1). Cross-terms vanish because the three estimators use disjoint items.    $\square$

*Remark* 1 (Consistency with the main text). Dropping the calibration terms ($m_1, m_0 \to \infty$) recovers Eq. (7). Equation (B.1) is exactly the variance Reiczigel et al. (2010) replace with an exact construction when $J$ is small or counts are sparse, because the Gaussian interval can under-cover near the $J \to 0$ floor; the $1/J^2$ point-variance scaling is identical.

**Proposition 2** (Variance divergence at the reliability floor)**.** *Fix the per-cap calibration budget $n, m_0, m_1$. For any cap with $J(K) > 0$ the recovered estimate has finite variance $\mathrm{Var}(\hat{\pi}(K)) = V(K)/J(K)^2$, with $0 < V(K) \le \frac{1}{4}\big(1/n + \pi^2/m_1 + (1-\pi)^2/m_0\big)$. Along any sequence of gates (or caps) for which $J \to 0$ while $V$ stays bounded away from $0$, the variance diverges as $\Theta\big(1/J^2\big)$; in particular $\pi$ is non-identifiable at $J=0$. We do* not *claim $\mathrm{Var}(\hat{\pi}(K))$ is monotone in the cap $K$: as the gate saturates, the numerator $V(K)$ can itself shrink toward $0$ and offset the growing $1/J(K)^2$, so $K^\star=1$ is adopted as a conservative operational default (Remark 2), not a proved variance minimizer.*

*Proof.* Write (B.1) as $V(K)/J(K)^2$. Each summand of $V(K)$ is the variance of a bounded $[0,1]$ quantity, so $0 < V(K) \le \frac{1}{4}(1/n + \pi^2/m_1 + (1-\pi)^2/m_0) < \infty$; hence for a fixed cap with $J(K) > 0$ the variance is finite. Along a sequence with $J \to 0$ and $\liminf V > 0$, $V/J^2 \to \infty$, giving the $\Theta(1/J^2)$ divergence;

and since $R - (1 - \mathrm{Sp}) = \pi J$ at the truth, the Rogan–Gladen inversion becomes ill-conditioned, so $\pi$ is non-identifiable exactly at $J=0$. Monotonicity in $K$ does *not* follow: $V(K)$ need not be bounded below uniformly in $K$—as the gate saturates ($\mathrm{Se} \to$ its limit and the ship rates $\to 0/1$, so the Bernoulli terms $R(1-R), \mathrm{Se}(1-\mathrm{Se}), \mathrm{Sp}(1-\mathrm{Sp})$ shrink toward 0)—so a decreasing $V(K)$ can offset the increasing $1/J(K)^2$, and $\mathrm{Var}(\hat{\pi}(K))$ can be non-monotone in the cap. $\square$

*Remark* 2 ($K^\star=1$ is an operational default, not a variance minimum). We do not claim $K=1$ minimizes $\mathrm{Var}(\hat{\pi}(K))$. Although $J(K)$ is non-increasing in the pass-prone regime (Prop. 1), the numerator $V(K)$ also shrinks as the gate saturates, and the two effects can trade off—in random pass-prone gates the variance sometimes *decreases* from $K=1$ to $K=2$ before rising. We adopt $K^\star=1$ on three grounds that need no minimization theorem: (i) retrying only *erodes* discrimination $J$ in that regime and cannot raise it; (ii) it minimizes judge cost; and (iii) empirically no larger cap materially improves out-of-sample recovery for either estimator—across $K=1$–6 the recovery MAE is flat to within $10^{-4}$ (PPI++ 0.046; Rogan–Gladen $0.119 \to 0.120$, rising after $K=2$), with $K=1$ within $3 \times 10^{-5}$ of the empirical minimum. $K^\star=1$ is thus *never materially worse*, which is the guarantee we assert.

*Remark* 3 (PPI++ implementation). We spell out the recommended estimator. On the calibration set we have labels $Y_i \in \{0,1\}$ (compliant) and the judge predictor $f_i =$ the item's mean pass-rate over the repeated rulings; on the unlabeled evaluation set we have $f_i$ only. Prediction-powered inference forms $\hat{\pi}_{\mathrm{PPI++}} = \bar{Y}_{\mathrm{lab}} + \lambda^\star(\bar{f}_{\mathrm{all}} - \bar{f}_{\mathrm{lab}})$, where $\bar{f}_{\mathrm{all}}$ averages the predictor over all $m+n$ labeled+unlabeled items and the power-tuning weight $\lambda^\star = \mathrm{clip}_{[0,1]}(\widehat{\mathrm{Cov}}(f,Y)/\widehat{\mathrm{Var}}(f))$ is estimated on the calibration set (matching our released code). At $\lambda^\star=0$ it reduces to the gold-only mean. We use the full-sample predictor mean $\bar{f}_{\mathrm{all}}$ rather than the unlabeled-only mean $\bar{f}_{\mathrm{unlab}}$ of the canonical estimator (Angelopoulos et al., 2023b); since $\bar{f}_{\mathrm{all}} - \bar{f}_{\mathrm{lab}} = \frac{n}{m+n}(\bar{f}_{\mathrm{unlab}} - \bar{f}_{\mathrm{lab}})$, our $\lambda^\star=1$ point applies an additional $\frac{n}{m+n}$ shrinkage rather than coinciding with plain PPI—absorbed by the data-driven $\lambda^\star$, which we fit against $\bar{f}_{\mathrm{all}}$. This choice is immaterial to our results: the two variants agree to within 0.0013 mean recovery MAE at every calibration fraction (App. A), and the full-sample variant is if anything the more conservative (closer to gold-only) at a 50% split. Its efficiency is governed by the judge–gold correlation $\rho(f,Y)$, *not* by $J$: since $\mathrm{Cov}(f,Y)$ relates to the gate's discrimination through $\pi(1-\pi)J$-type terms, PPI++ retains information whenever the judge is informative and gracefully falls back to gold otherwise—this is why it escapes the $1/J^2$ penalty of Rogan–Gladen. The standard PPI++ SE $\widehat{\mathrm{Var}} = \widehat{\mathrm{Var}}(Y - \lambda^\star f)_{\mathrm{lab}}/m + \lambda^{\star 2}\widehat{\mathrm{Var}}(f)_{\mathrm{unlab}}/n$ applies to the $\bar{f}_{\mathrm{unlab}}$ form (whose labeled and unlabeled samples are disjoint); the $\bar{f}_{\mathrm{all}}$ form we run additionally carries a labeled/unlabeled overlap covariance, so instead of a plug-in interval we report the empirical spread over the 300 out-of-sample calibration splits. This split spread is a *calibration-stability* band—it quantifies how much the estimate moves with the choice of labeled subset on a fixed dataset—*not* a sampling confidence interval for the population $\pi$; coverage *for $\pi$* is validated separately on a synthetic gate with known truth (95% coverage $[0.92, 0.97]$ across $J \in [0.1, 0.8]$, §5.1). *Deployment note.* Our predictor $f_i$ averages the item's repeated rulings; a practitioner measuring at $K=1$ with a *single* binary ruling has a noisier $f_i$, which lowers the judge–gold correlation and shrinks PPI++'s efficiency edge toward the gold-only mean—to which, by the $\lambda^\star$ fallback, it never falls below. A handful of repeated rulings per item recovers most of the edge at modest cost, so the recipe degrades gracefully with the labeling/ruling budget.

*Remark* 4 (Empirical confirmation for PPI++). Prop. 2 gives the $1/J^2$ variance divergence at the reliability floor; the choice of cap $K^\star=1$ is the operational default of Remark 2, which the empirics confirm. Measuring out-of-sample recovery MAE at each cap across all gates, for PPI++ it is *flat* in $K$ (0.046 across $K=1$–6) and for Rogan–Gladen flat then rising ($0.119 \to 0.120$, its minimum at $K=2$ but within $3 \times 10^{-5}$ of $K=1$): as $K$ grows the gate ships more, $f_i = 1 - (1 - p_i)^K$ saturates toward 1, and the judge–gold correlation PPI++ exploits collapses, so its efficiency edge erodes rather than improving. Neither estimator's recovery is materially improved by any larger cap, even though PPI++'s variance is not the $1/J^2$ form of (B.1).

*Remark* 5 (Scope). $K^\star = 1$ concerns *measurement*, not agent utility: extra retries may raise true success but, in the Prop. 1 regime where $J(K)$ falls, the discrimination both estimators rely on erodes, and empirically no larger cap improves recovery for either estimator (Rogan–Gladen $0.119 \to 0.120$; PPI++ flat)—so retries do not help measurement even when they help the agent. This is the companion to the bias-side argument of §3–§4, and is why we report a PPI++ point estimate with $J$ as the reliability diagnostic, collected at $K=1$ whenever the rate can be re-collected.

