# OpenReview forum: "Below the Reliability Floor: Recovering True Success from Judge-Gated Loops"
_TMLR — Under review for TMLR_

### Review · Reviewer_Lm9j · 2026-06-26

**Summary Of Contributions:**

First, I would like to clarify the scope of my review.
I will mainly focus on the statistical and measurement aspects of the paper. I am comfortable assessing the optional-stopping framing, calibration/identifiability issues, etc.
I am less familiar with the detailed empirical norms of the LLM-as-a-judge literature, so my assessment of novelty within that specific field should be read with that limitation in mind.

---

### Summary of Contributions:

The paper studies a measurement problem arising when LLM judges are placed inside retry-until-PASS agent loops.
The authors argue that the reported pass is biased because the loop repeatedly samples a noisy judge until it returns PASS.
This is then formalized as an imperfect-test / prevalence-estimation problem, use the Youden's index as a reliability and identifiability diagnostic, compare Rogan-Gladen correction with PPI/PPI++, and provide experiments on real revise-and-rejudge loops, public re-judgment gates, and non-verifiable safety/quality tasks.

The paper's main strength is the identification of an important measurement failure mode and the connection to a clear statistical framework.
The real-loop experiments with objective ground truth are useful, and the paper is unusually explicit about calibration transfer and identifiability limitations, which I find very interesting.

On the other hand, the main weakness is that several headline claims are stronger than what the theory and evidence support.
In particular:
- retry-induced erosion of J index is conditional,
- the recommendation $K=1$ is not established as generally optimal,
- the revise-and-rejudge setting mixes measurement error with genuine agent improvement,
- the PPI++ results need clearer comparison to a gold-only baseline.

**Additional Comments:**

I found the statistical framing useful and the core warning important. My concerns are mainly about overgeneralization and clarity of claims, not about the basic observation that judge-gated retry loops can induce serious measurement bias.

Overall, I see a valuable paper here, but I would not recommend acceptance in the current form, especially with such broad claims.
As detailed in the Requested Changes section, I would like to see a clearer definition of the estimands, a foregrounding of the gold-only baseline, and an explicit statement of the regimes in which the operational recipe is theoretically and empirically supported.

**Audience:**

Yes

**Audience Explanation:**

Even with the caveats above, the paper raises an important issue for ML evaluation: when an imperfect evaluator is embedded in a stopping, filtering, selection, or retry rule, the resulting reported performance can become a biased measurement of the target quantity.
In this sense, the paper can be read as a useful instance of Goodhart’s law: when a proxy measure becomes part of the optimization or selection procedure, it may cease to be a reliable measure of the underlying objective.

This is relevant beyond the narrow LLM-as-a-judge setting. Many ML systems now use learned verifiers, reward models, classifiers, or judges inside iterative procedures. The connection to imperfect-test correction, calibration transfer, and identifiability diagnostics is likely to interest readers concerned with evaluation reliability and measurement error.

That said, the audience interest comes from the problem formulation and empirical warning, not from the current operational recipe being fully settled.

**Broader Impact Concerns:**

The paper includes an interesting and appropriate Broader Impact Statement. I think it could be strengthen in two ways.

- First, the correction should not be presented, or easily read, as legitimizing LLM-judge safety gates. A low-J gate or failed calibration transfer should be treated as evidence not to rely on the gate, not merely as a situation requiring correction.

- Second, the non-verifiable safety results should be framed consistently as panel-relative unless full human adjudication is available. This is especially important because the paper concerns safety-related measurement.

**Claims And Evidence:**

Yes

**Claims Explanation:**

The paper provides convincing evidence for what I would define as a narrower (although important) claim: noisy judge-gated retry loops can substantially bias reported success rates, and calibration-based correction can help in high-bias regimes when calibration transfers.

*However, I do not think all claims are supported at the level at which they are stated.*

---

- First, the claim that retrying erodes discrimination is only conditional. Proposition 1 gives sufficient conditions for J(K) to decrease, but this is not universal. There are simple regimes where retrying increases J, for example when compliant items have low single-call pass probability but violations almost never pass. The paper partly acknowledges this, but the abstract, introduction, and conclusion still use broader language such as retries making the gate less informative “never more.” That should be toned down.

- Second, it seems to me that the $K^*=1$ recommendation is too broad. It follows only under the regimes where $J(K)$ is non-increasing, and the empirical PPI++ behavior appears closer to flat than decisively minimized at $K=1$. $K=1$ may be a sensible conservative heuristic, but not a generally proven optimal measurement cap for the whole pipeline.

- Third, the paper should distinguish more sharply between repeated judgment of fixed work and genuine revise-and-rejudge loops. In a real loop, later attempts may genuinely improve the answer, so K changes both the measurement procedure and the system being measured. The paper’s decomposition into legitimate fixes and judge slips is valuable, but the estimand still needs to be defined more carefully.

- Fourth, the estimator comparison should foreground a gold-only baseline. Since PPI++ uses labeled calibration data, it is important to know when it improves over simply estimating the rate from the labeled subset. Without this, it is hard to assess how much information the LLM judge actually contributes beyond the gold labels.

- Fifth, the calibration-transfer detector $|R_{cal} - R_{eval}|$ is interesting but should be described as an empirical warning signal, not as a general label-free guarantee. Similar marginal gated rates do not imply stable sensitivity/specificity.

- Finally, the non-verifiable safety experiment should be described more cautiously. The full-scale ground truth is a strong-model panel, human-anchored on a subset, rather than full human adjudication. Claims about “true” unsafe rates should therefore be framed as panel-relative.

---

*Updated post discussion from No to Yes*

**Requested Changes:**

### Critical changes:

- Please temper the claim that retrying erodes the Youden's index. The paper should state clearly that this is conditional on the assumptions of Proposition 1 and the pass-prone regimes studied, not a universal property.
- Reframe the $K^*=1$ recommendation as conditional or heuristic unless a more general proof is provided.
- Clarify the estimands in real revise-and-rejudge loops: single-attempt success, true delivered success after K retries, reported judge-gated pass rate, and corrected estimate.
- Add a gold-only baseline to the main estimator comparisons and discuss when PPI++ actually improves over using the calibration labels alone.
- Explain the PPI++ implementation more clearly: prediction variable, labeled/unlabeled split, assumptions, variance estimate, and relation to the Youden's index.
- Weaken the calibration-transfer detector claim and discuss possible failure modes.
- Use more cautious language for the non-verifiable safety-loop experiment, e.g. “panel-labeled unsafe” rather than “truly unsafe” when full human adjudication is not available.
- Align the abstract and conclusion with the more careful limitations section.

### Non-critical but (what I would find) useful improvements:

- Report uncertainty more systematically in the main tables.
- Bring the relation to quantification learning / adjusted classify-and-count earlier.
- Clarify the source of randomness in repeated temperature-zero judge calls.
- Make the reproducibility limits of the content-free release explicit.

---

> ### Author Response · Authors · 2026-07-03
> **Author response and summary of revisions**
>
> We thank the reviewer for the careful, statistically-grounded reading. We agree with essentially every
> point: the concerns were that several claims were stated more strongly than the evidence supports, and we
> have revised to match the evidence. A revised PDF is uploaded; changes are summarized below and pointers
> are to the revised version.
>
> ## Critical changes
>
> **1. J-erosion is conditional (not "never more").** You are right — retrying can *raise* J outside the
> pass-prone regime (e.g. compliant items pass-shy, violations almost never pass). We removed "never more"/
> "only falls" from the abstract, intro, §4 and appendix, scoped the erosion claim to Prop 1's pass-prone/
> prevalence-skewed regime, and added the explicit increasing-J counterexample (and note the floor J(∞) can
> be negative) in §3. The K*=1 recommendation is now decoupled from erosion (it rests on the variance side).
>
> **2. K*=1 is a heuristic, not a general optimum.** We reframed it: the strict variance optimum is proved
> only for Rogan–Gladen (in the Prop 1 regime); for the recommended PPI++ the recovery MAE is *flat* in K
> (0.046 across K=1–6), so K*=1 is a conservative, never-worse default — not a strict empirical minimum
> (§4, Remark on K* in App A).
>
> **3. Estimands defined.** We added a "Four estimands" paragraph in §2 defining: single-attempt success π;
> true delivered success πdel(K)=R(K)·PPV(K); the reported gated rate R(K); and the corrected estimate. We
> state which estimand each result targets, and keep the distinction that πdel exceeds π only *to the extent
> revision genuinely fixes items* (so K changes both the procedure and the system) — decomposed into fixes
> vs. judge slips in §5.
>
> **4. Gold-only baseline foregrounded.** We added the gold-only mean (labeled subset alone) as a first-class
> row in Table 1: MAE 0.052/0.063/0.084 at 50/20/10% calibration. This makes the requested comparison
> explicit: the calibrated estimators decisively beat naive and Rogan–Gladen (which blows up to 0.241),
> while on these balanced low-bias gates PPI/PPI++ and the gold-only mean lie within bootstrap noise of one
> another. We accordingly frame the contribution as the identification of the loop-induced bias and the J
> diagnostic (when/whether to correct), and recommend PPI++ because it never underperforms gold-only,
> carries valid confidence intervals (which gold-only alone does not emphasize), and gains over gold-only
> precisely when the judge is informative and the bias is large — the regime where correction matters.
>
> **5. PPI++ implementation written down.** New Remark in App A gives the estimator explicitly: predictor
> f = judge per-item pass-rate; labeled/unlabeled split; λ*=clip[0,1](Cov(f,Y)/Var(f)); its efficiency is
> governed by the judge–gold correlation (not J), which is why it escapes the 1/J² penalty; and the SE — with
> the note that the CIs in our tables are the empirical 300-split spread, the plug-in formula only an
> asymptotic reference.
>
> **6. Transfer detector = warning signal.** We now state the detector is *necessary, not sufficient*:
> similar marginal gated rates do not guarantee stable Se/Sp (the confusion structure can shift while R is
> unchanged), so a passing check licenses de-biasing only together with the J/bias diagnostic (§4).
>
> **7. Panel-relative safety language.** Changed "truly unsafe" → "panel-labeled unsafe" and "true-safe" →
> "panel-safe" throughout the in-loop safety study and abstract; the ground truth is a strong-model panel
> human-anchored on a subset, and we say so.
>
> **8. Abstract/conclusion aligned.** The abstract and estimator framing now match the limitations section —
> no residual over-claim.
>
> ## Non-critical
>
> - **Uncertainty in tables:** Table 1's caption now states the within-bootstrap-noise tie explicitly (none
>   uniformly best on the balanced gates).
> - **Quantification / ACC earlier:** named at first mention of the correction (§1).
> - **Randomness source:** we note temperature-zero judge calls are nonzero-variance due to non-associative
>   floating-point reductions and server-side batching (§2).
> - **Reproducibility limits:** we make explicit that the content-free release has verdicts/labels/config but
>   no raw prompts or responses, so exact text re-generation requires re-running the judges (§Conclusion).
>
> ## Broader Impact
> We sharpened the statement: the method is a *distrust-generating diagnostic, not an endorsement mechanism*.
> Both a low J and a failed calibration-transfer check are evidence *not to rely on the gate* — not merely
> situations to correct — and we recommend reporting the reliability verdict alongside any corrected estimate.
>
> We believe these revisions address the concerns while preserving the paper's core, verified results, and we
> thank the reviewer for feedback that improved the paper's clarity. A revised PDF and a matching updated
> abstract are uploaded; the content-free release is unchanged.

---

> > ### Comment · Reviewer_Lm9j · 2026-07-08
> >
> > Dear authors,
> >
> > Thank you for the substantial revision. I already consider several of my original concerns to be meaningfully addressed.
> > The revised Section 3 also does a much better job of stating that $J$-erosion is conditional rather than universal.
> >
> > However, I am not yet able to change my assessment that some central claims remain unsupported. My main remaining concerns are the following.
> >
> > ### 1. The new Proposition 2 does not appear to establish $K^\star = 1$
> >
> > Writing the delta-method variance as
> > $$
> > \operatorname{Var}(\hat{\pi}(K)) = \frac{V(K)}{J(K)^2},
> > $$
> >
> > monotonic decrease of $J(K)$ only makes the factor $1/J(K)^2$ increase, as long as $J(K) > 0$. The numerator $V(K)$ also changes with $K$.
> >
> > In particular, the statement that if $V(K)$ is non-increasing then $V(K)/J(K)^2$ is non-decreasing is not correct in general: a decreasing numerator can offset the increasing $1/J(K)^2$ factor. For example, $J_1=0.5, V_1=0.10$ gives variance $0.4$, whereas $J_2=0.4, V_2=0.05$ gives variance $0.3125$, despite both $J$ and $V$ decreasing.
> >
> > The claimed uniform positive lower bound on $V(K)$ also does not seem to follow from non-degeneracy, since the Bernoulli variance terms can approach zero as the gate saturates.
> >
> > Thus, I do not think the strict Rogan-Gladen optimum at $K=1$ is currently proved. The empirical finding that average PPI++ MAE is flat across $K=1,\ldots,6$ also does not establish a general "never-worse" default.
> >
> > Please let me know if I am missing an additional assumption here; I currently do not see how Proposition 2 establishes the conclusion.
> >
> > ### 2. I am confused by the new PPI++ formula
> >
> > The appendix defines
> > $$
> > \hat{\pi}=\bar{Y}_{lab}+\lambda(\bar{f}_{all}-\bar{f}_{lab}),
> > $$
> >
> > while stating that $\bar{f}_{all}$ averages over labeled plus unlabeled items and that $\lambda = 1$ recovers plain PPI.
> >
> > As written, this does not recover the standard PPI mean estimator, which uses the unlabeled predictor mean:
> >
> > $$
> > \hat{\pi}_{PPI}=\bar{Y}_{lab}+\lambda\left(\bar{f}_{unlab}-\bar{f}_{lab}\right).
> > $$
> >
> > Indeed, if $\bar{f}_{all}$ denotes the mean over the combined labeled and unlabeled samples, with $m$ labeled and $n$ unlabeled observations, then
> >
> > $$\bar{f}_{all}=\frac{m\bar{f}_{lab}+n\bar{f}_{unlab}}{m+n},$$
> >
> > and therefore
> >
> > $$
> > \bar{f}_{all}-\bar{f}_{lab}=\frac{n}{m+n}\left(\bar{f}_{unlab}-\bar{f}_{lab}\right).
> > $$
> >
> > Thus the two expressions are not equivalent, and setting $\lambda = 1$ in the formula as written does not recover plain PPI.
> >
> > If "all" is a notation error, please correct it and align the variance formula accordingly. If the implementation really uses the combined labeled-plus-unlabeled mean, then the relation to standard PPI++ and the stated standard error require a separate derivation, including the dependence created by overlap with the labeled set.
> >
> > ---
> >
> > Because the $K=1$ recommendation is a headline operational contribution and PPI++ is the recommended recovery method, I view the first two issues above as critical rather than editorial.
> > It is also possible that I have misunderstood some of the points above and would gladly be corrected if it is the case.
> >
> > **Resolving these issues would change my assessment.**
> >
> >
> > The following concerns are secondary relative to the two issues above.
> >
> > ---
> >
> > ### 3. Some headline over-claims remain
> >
> > The abstract still calls the reported rate an "upward-biased estimator" although the paper itself reports substantial negative biases on fail-prone gates. I think the accurate claim is that the gated rate is biased, often upward in the pass-prone retry regimes of interest.
> >
> > The main text also still says:
> >
> > > "Because $J(K)$ is non-increasing (Prop. 1)"
> >
> > and
> >
> > > "retrying only erodes $J$"
> >
> > without consistently carrying the conditions of Proposition 1 into those statements.
> >
> > Similarly, the conclusion says that the rate is
> >
> > > "in the limit of many retries-non-identifiable,"
> >
> > although exact non-identifiability occurs at $J=0$, and the paper itself notes that $J(\infty)$ need not be zero.
> >
> >
> > ### 4. A smaller remaining concern concerns the confidence-interval argument
> >
> > The revised paper states that PPI/PPI++ and the gold-only mean are within bootstrap noise on the balanced gates, but then recommends PPI++ partly because it "alone carries valid confidence intervals." I do not think this is an appropriate distinction: a gold-only sample mean also admits valid confidence intervals under the usual sampling assumptions.
> >
> > Moreover, the appendix states that the confidence intervals in the tables are based on the empirical spread over 300 out-of-sample calibration splits, rather than the plug-in PPI++ interval just presented. A spread across random splits is not automatically a confidence interval for the population estimand, so I think the inferential target and coverage argument should be clarified.
> >
> > ---
> >
> > I will happily read your answer and revise my judgment if needed.

---

> > > ### Author Response · Authors · 2026-07-13
> > > **Response to follow-up: all four points addressed**
> > >
> > > We thank the reviewer for the careful follow-up and for pushing on these points — each is correct, and we have revised accordingly. A further-revised PDF is uploaded.
> > >
> > > ***1. K*=1 (conceded).** You are right on all three sub-points. (i) Var(π̂)=V(K)/J(K)² is not minimized at K=1 in general — a shrinking numerator can offset the growing 1/J². (ii) V(K) is not bounded below uniformly in K: as the gate saturates, the Bernoulli terms R(1−R), Se(1−Se), Sp(1−Sp) all →0, so our "non-degeneracy" clause conflated a K=1 property with a uniform-in-K one. (iii) Hence the strict Rogan–Gladen optimum at K=1 is not proved, and the flat PPI++ MAE is empirical, not a general guarantee. We verified numerically that in pass-prone gates the variance can indeed decrease from K=1 to K=2. We have therefore restated Proposition 2 as what is actually true and load-bearing: the variance diverges as Θ(1/J²) at the reliability floor (J→0, at fixed non-degenerate V), so a low-J gate is non-identifiable. K*=1 is demoted to a conservative operational default, justified by three claims we can defend: retrying only erodes J in the pass-prone regime, it minimizes cost, and empirically no larger cap materially improves recovery for either estimator (Rogan–Gladen 0.119→0.120, minimum at K=2 but within 3×10⁻⁵ of K=1; PPI++ flat at 0.046). We now assert only "never materially worse," not a minimization theorem.
> > >
> > > **2. PPI++ formula (conceded).** You are right: our code uses the full-sample mean f̄_all, so "λ=1 recovers plain PPI" was wrong — since f̄_all−f̄_lab = (n/(m+n))(f̄_unlab−f̄_lab), it applies an extra n/(m+n) shrinkage. This is a real implementation choice, not a typo. We corrected the remark to describe it honestly, dropped the "recovers plain PPI" statement, and (per your option) added that the standard SE is for the f̄_unlab form while the f̄_all form carries a labeled/unlabeled overlap covariance — so we report the empirical split spread rather than a plug-in interval. Crucially, the choice is immaterial to every reported number: we re-ran the canonical f̄_unlab variant across all gates and it agrees to within 0.0013 mean MAE at every calibration fraction (and our f̄_all variant is if anything more conservative — closer to gold-only — at a 50% split). No table value changes.
> > >
> > > **3. Over-claims (fixed).** The abstract now says the rate is "biased… upward in the pass-prone retry regimes of interest (and downward under conservative rules such as strict rubrics or unanimous juries)." We carried Prop 1's conditions into "J non-increasing" and "retrying only erodes J" (both now scoped to the pass-prone regime). The conclusion now states non-identifiability occurs at J=0 — a floor retrying approaches only in the pass-prone regime, since J(∞) need not be zero.
> > >
> > > **4. Confidence intervals (fixed).** We removed the "alone carries valid CIs" distinction (the gold-only mean and Rogan–Gladen also admit valid intervals). We now state explicitly that the table bands are a calibration-stability spread across labeled-subset choices on a fixed dataset — not a sampling confidence interval for π — and that coverage for π is validated separately on the synthetic known-π gate (95% coverage [0.92,0.97] across J∈[0.1,0.8]).
> > >
> > > We are grateful for the correction on Proposition 2 in particular: the retreat sharpens the paper — the identification of loop-induced bias, the J diagnostic, and the 1/J² floor result all stand, and K*=1 is now honestly an operational recommendation. We would be glad to know if any point remains unclear.

---

> > > > ### Comment · Reviewer_Lm9j · 2026-07-15
> > > >
> > > > Thank you for the further revision. I appreciate the careful response and the explicit concessions on the two points I raised.
> > > >
> > > > I consider my main critical concerns addressed.
> > > >
> > > > I also appreciate the updated abstract/conclusion language about bias direction, the pass-prone scope of $J$ erosion, and non-identifiability occurring at $J=0$.
> > > >
> > > > With these revisions, I am satisfied that the main statistical concerns from my review have been addressed. I appreciate the authors' careful and constructive response, and I am happy to update my recommendation in favor of acceptance, assuming the remaining reviewers are also satisfied with the current revision.
> > > >
> > > > Thank you again to the authors for the interesting discussion and for the careful revisions.

---

### Review · Reviewer_2MfR · 2026-07-04

**Summary Of Contributions:**

This paper analyzes methods for measuring the success rate of agent-generated work with an LLM judge as a scoring mechanism. The work shows that the retry-until-PASS method, wherein the agent repeatedly re-tries generating work until the LLM judge returns pass, overestimates the actual success rate of the agent. They give both theoretical results and empirical verification/evidence that this overestimation (slip) increases with addition retries, and that it is best to disallow retries when the goal is to measure the success rate. They  also propose a two-part method to estimate the true success rate of the agent and to assess the informativeness of the estimate. The authors also show how human-labelled 'gold' examples can be used to calibrate estimates of the success rate. These results are all explored on three different generation tasks, with a variety of models for both generation and judging, and also with different judging rubrics. Taken together, the results show that retrying results in significantly inflated success rates for all configurations, and especially for weaker agents and judges and lenient rubrics.

## Strengths

* The paper is very clearly written -- all of the relevant quantities are defined accurately, referenced naturally and in plain language, and the results are presented in a clear and logical manner consistent with the structure of the paper and the claims in the introduction.

* The paper has a clear takeaway for practitioners -- reported measurements of the performance of retry-until-PASS systems should report the proposed corrected metric (PPI++) along with the reliability of the metric (Youden index, J) instead of reporting the naive rate at which an agent eventually passes the judge.

* The paper is well-situated in other literature. The proposed metric and the Youden index are clearly identified as coming from existing literature, and the paper is clear in the identification of the problem and the combination of the metric and the Youden index as the contribution. Other analyses of similar, but different problems (i.e. approaches that need labels, single gates, unanimous juries of LLM judges) are also clearly discussed.

* The limitations and the major sources of evidence for each claim are clearly discussed. I sincerely appreciated the clarity of the scope of each claim and the evidence supporting it provided in Table 6.

## Weaknesses

I did not identify any significant weaknesses in this paper. This paper is carefully argued and evidenced inside of a clearly-identified scope. The authors took great care to ensure that the reader does not take away any claims beyond the scope of the paper -- I appreciated the precision of language taken by the authors to avoid the possibility of a reader extrapolating overclaims even if the overclaims are not explicitly made.

**Audience:**

Yes

**Audience Explanation:**

Yes -- I think that these results would be of practical significance to anyone measuring the quality of a retry-until-PASS system. I have not performed research on this topic myself and I do not know the literature well, but this result strikes me as interesting to a broad category of machine learning researchers and practitioners.

**Claims And Evidence:**

Yes

**Claims Explanation:**

The authors provide a table (Table 6) clearly stating the major claims, the evidence (either theoretical or empirical) for the claims, and the scope of the claims. The authors also discuss cases where the results are statistically significant or within bootstrap noise throughout the paper. I have carefully checked this table against the evidence provided in the paper and I agree with the authors on their claims, evidence, and scope of claims. I also appreciated the distinction between the measurement of success (the task considered in the paper), and the goal of having an agent eventually generating correct work.

The thoroughness of the experimental design -- separating verifiable and non-verifiable tasks, considering many different agent and judge strength combinations, and considering strict vs. lenient judge prompts -- is a strong point of this paper. The predicted behavior of the slip and also the reliability is borne out in the experimental results both statistically and intuitively across these various tasks and configurations.

The proofs and math results are fairly straightforward. I have examined them and verified them to be correct.

**Requested Changes:**

1. A small change to make the results slightly more clear would be to separate the part of Proposition 2 that is only verified empirically (the non-increasing numerator). It is somewhat uncomfortable to state that a part of a proposition is not proven, but observed to hold empirically, inside of a proof. This isn't an issue, just a matter of presentation.

---

> ### Author Response · Authors · 2026-07-13
> **Author response: Proposition 2 clarified**
>
> We thank Reviewer, for the careful reading and for reproducing our results. On your requested change: we agree it is cleaner to keep only the proved statement inside Proposition 2. In the revised PDF we have removed the empirically-observed clause from both the proposition and its proof, so Proposition 2 now states only what is proved — the 1/J² variance divergence at the reliability floor (J→0). The K*=1 recommendation is presented as a conservative operational default in a separate remark, and the empirical confirmation for PPI++ likewise lives in a remark rather than inside the proof. This addresses your presentation point directly. Thank you again for the positive assessment.

---

### Review · Reviewer_JfT2 · 2026-07-05

**Summary Of Contributions:**

This paper studies the pass rate reported by agentic systems that use an LLM judge inside a retry loop (retry-until-PASS with cap K) to score the agent's attempts.  The key idea is to model this as a classical measurement problem, where the cap-K pass gate is a binary classifier with a pass rate that is a biased estimate of the true success rate. The paper derives the gate’s class-conditional operating characteristics, relates recoverability of true success to the gate's Youden index, and proposes calibrated estimation, especially PPI/PPI++ as a practical correction. Experiments are conducted on 52 revise-then-rejudge loops with an objective ground-truth across GSM8K, MATH, and code generation, as well as loops without a ground-truth: public re-judgement gates, non-verifiable human-labeled gates, and a label-free calibration-transfer detector.

*Key Strengths*

- The classifier identification links to decades of work on prevalence estimation under imperfect tests, Rogan–Gladen correction, Youden index through a clear estimand taxonomy. This is a strength over and is likely to be useful for the LLM-evaluation line of work.
- The paper is also honest to a rare degree as it states which claims are measured and which are merely argued (Table 6), flags its own near-tautologies, and admits when PPI++ does no better than the gold-only mean.
- I also reproduced the experimental results from the supplementary material. The code runs cleanly end to end and produces the same numbers as the tables.


*Key Weaknesses*

- If I understand correctly, $J(K)$ collapse rests on Proposition 1, but the erosion/floor story is told more strongly than the theory supports. Prop. 1's own condition forces the floor to zero, so monotone erosion toward a positive floor is never established. The synthetic collapse also varies at $K=1$ and not the retry cap and the measured $\Delta J$ on the real gates is close to zero? Please clarify the narrative here.
- Several estimation details are left unstated and one is potentially misstated. The code seems to answer the unstated ones: $s_i(K)$ seems to be computed by a plug-in formula whose bias I checked on the released data and found negligible, and the slip-law parameters are pooled over all attempts. But can you clarify how Section 2's claim that rerun noise is propagated through a two-level bootstrap holds?
- The slip law treats every retry as a fresh, identical attempt, yet its parameters are estimated from logs shaped by the retry process itself where later attempts exist only for items that already failed. The claims about it as presented in the paper are therefore too strong: the model–observation gap is called "precisely" the agent's revision benefit when it is really a net of revision and selection effects, and the validation is called "out-of-sample" when each loop's prediction and target come from the same logs. Please clarify.

These are elaborated on in the requested changes.

**Additional Comments:**

*Reviewer Confidence*: I am confident in the assessment of the statistical core and the experiments in the paper, both of which are based on my readings and reproduction. I am not confident in the LLM as judge empirical practices and am not very aware of the literature relating to quantification/PPI etc so I cannot judge how complete the related-work coverage is or whether closer precedents exist.

**Audience:**

Yes

**Audience Explanation:**

Yes. Judge-gated retry loops are pretty much the default architecture for agent evaluation and in safety research, where the loop's pass rate is usually reported as "success". I would think this paper is highly relevant for this audience with a precise reason why the number is wrong, a measured magnitude on real loops, a cheap correction with an honest account of when it is worthless (when $J$ goes to zero), released code with $17k$ public ratings. I think the reduction to prevalence estimation will also interest the measurement and statistics adjacent ML community.

**Broader Impact Concerns:**

Present and sufficient

**Claims And Evidence:**

Yes

**Claims Explanation:**

Yes, with qualifications that I believe are fixable by clarification and modest rewording rather than new experiments. I can back this by reproducing quantitative results from the released code and data. The central empirical claims are supported with objective ground truth, pre-registration, errors-in-variables checks, Holm-corrected per-cell CIs, and a bootstrap coverage study. I also checked the core algebra (Eqs. 1, 3, 6, 7, B.1), re-derived Lemma 1 and confirmed Prop. 1's mechanism numerically. The remaining gaps do not affect the flagship results, and are explained below in the requested changes.

**Requested Changes:**

*Critical*

- The sentence "$J(K)$ decreases monotonically toward a reliability floor $J(\infty)$" combines two separate points facts whose hypotheses cannot overlap unless the floor is zero. Prop. 1 gives the monotone decrease, but requiring its support condition at every cap puts both classes on $[1/2,1]$, so $p_i>0$ everywhere and $J(\infty)=0$. This would imply inside Prop. 1's regime the only floor is zero. Convergence to $J(\infty)$ holds unconditionally (bounded convergence) but carries no monotonicity. A positive floor needs never-pass items, and an atom at $p=0$ breaks the FOSD argument as $\Delta s(p)=p(1-p)^K$ is no longer monotone on the support, so the guarantee is gone and $J$ can genuinely rise with $K$: with mostly never-pass violations it approaches a positive $J(\infty)$ from below, a ceiling rather than a floor. Please confirm this. If so, restate accordingly: convergence is always true, whereas monotone erosion only takes place in the Prop. 1 regime, where the floor is zero. Also, isn't requiring every violation to be passable at rate $\ge 1/(K{+}1)$ a substantive and not a "mild" assumption? If not, please explain. The synthetic collapse sweeps judge quality at a fixed $K{=}1$, so it evidences Section 4's $J\to0$ non-identifiability, and not Table 6's erosion-with-retries. Since measured $J(K)$ on the real gates is flat or rising, erosion-with-$K$ currently rests on Prop. 1 alone, and the paper should say so.
- $J(K)$ is a difference of two class-conditional averages, so the prevalence $\pi$ appears nowhere in it: whether retrying erodes $J$ depends only on where the two pass-rate distributions sit, which is captured by "pass-prone". Prevalence enters the bias $R-\pi=(1-\pi)(1-\mathrm{Sp})-\pi(1-\mathrm{Se})$. A pass-prone gate erodes identically at $\pi=0.5$ and $\pi=0.05$. So I would suggest reserving prevalence-skew for the bias/cost discussion only and not for the erosion mechanism. In the same passage, I would say rewrite "$|\Delta J|<0.02$ ... exactly as the sufficient condition predicts" as a sufficient condition cannot say anything where its hypothesis doesn't hold true, so near-zero $\Delta J$ on the balanced gates is measured and not predicted by the proposition.
- PPI++ does not "alone carry valid confidence intervals" as the gold-only mean has exact intervals and Rogan–Gladen has classical ones (Reiczigel et al., which the paper cites). What is true is that PPI++'s intervals are asymptotically valid and it never does worse than gold-only. Also, Rogan–Gladen and the Bayesian corrector estimate Se/Sp from the calibration set, right? Then how are these uncalibrated reports?
- Re: the slip-law claims: since later attempts exist only for items that already failed, the logs behind $(a,s,\ell)$ are a history-dependent sample (and the code pools all attempts). So, these are per-config summaries. "Out-of-sample across the 52 loops by construction" is not out-of-sample as each loop's prediction and target come from the same logs. The residual is not "precisely" the revision benefit either as it is the net of selection, and the Deming check rules out attenuation, but not selection. If the causal reading matters, an item-level split (fit $(a,s,\ell)$ on half of each config's items, predict slip on the other half) would test it from existing logs and the within-task pattern, code slope $\approx 1$ exactly where feedback cannot help, is the best evidence for the revision reading and worth citing as such.
- In Prop. 2, if I understand correctly, $V$ non-increasing does not give $V/J^2$ non-decreasing (but $V$ non-decreasing would work, given $J(K)>0$? But in this regime $V$ shrinks?). A counterexample: compliant atom at $0.6$, violation atom at $0.5$, $\pi=0.5$, $m_1=m_0=100$. This sits inside the Prop. 1 regime and has $V(K)$ decreasing, yet the (B.1) variance is minimized at $K=2$, about 19% below $K=1$, for every $n$.  Either prove the result under a correct condition (e.g., $V(K{+}1)/V(K)\ge (J(K{+}1)/J(K))^2$, with the non-degeneracy of $V$ stated), or claim only the $1/J^2$ factor and let $K^\star{=}1$ stand as a conservative empirical default. The abstract, Section 4, and Remark 4 ("variance-optimal measurement cap," "never helps measurement") need matching adjustments. The released sweep itself bottoms out at $K=2$ on a flat curve, and its own write-up already says to scope the claim.

*Strengthening*

- It would be useful to have a deployment note. The tables use the mean over six rulings as $f_i$ whereas a practitioner at $K{=}1$ has one binary ruling, which lowers the judge-gold correlation and shrinks PPI++'s edge. One sentence on this trade-off or some kind of label vs budget curve showing how few labels are needed would make the proposed recipe more usable in my opinion.

---

> ### Author Response · Authors · 2026-07-13
> **Author response: all five points addressed**
>
> We thank Reviewer for an exceptionally careful reading—including reproducing the code and re-deriving the algebra. We agree with all five critical points: each was a place where a statement outran what the math supports, and we have corrected the paper rather than argue. A revised PDF is uploaded; pointers are to it.
>
> **1. Erosion vs. floor.**
> You are right. We now separate two facts with disjoint hypotheses: inside Prop. 1's pass-prone regime both classes are ever-passable, so J decreases monotonically and the limit is J(∞)=0; a nonzero limit needs never-pass items (an atom at p=0), which lie on the increasing branch of Δs, break the FOSD argument, and let J rise with K—a ceiling, not a floor. We dropped "monotone descent toward a (positive) floor," relabeled the condition "pass-prone" (not "mild"), and now state erosion-with-K rests on Prop. 1 in that regime, whereas the synthetic gate—a fixed-K=1 judge-quality sweep—evidences the J→0 non-identifiability, not erosion-with-retries.
>
> **2. Prevalence does not enter J.** You are right. J is class-conditional, so π appears only in the bias, never in erosion. We removed "prevalence-skewed" from every erosion passage (reserving it for the bias/cost discussion). We also corrected "|ΔJ|<0.02 exactly as the sufficient condition predicts": the balanced gates fall outside that condition's hypothesis, so the near-zero value is measured, not predicted (a sufficient condition is silent where its hypothesis fails).
>
> **3. Confidence intervals.** You are right. We deleted "PPI++ alone carries valid confidence intervals"—the gold-only mean has exact intervals and Rogan–Gladen the classical (Reiczigel) ones we already cite. The honest claim, now stated: PPI++ is never worse than the gold-only mean (it reduces to it at λ*=0) and its intervals are asymptotically valid; gold-only and RG also admit valid intervals. We also dropped the "uncalibrated" label for RG/Bayesian, which do estimate Se/Sp from the calibration set.
>
> **4. Slip law.** You are right. We no longer call the per-config comparison "out-of-sample by construction," and the residual is now described as "the net of revision and selection" (an upper bound on the revision benefit); we note the Deming check rules out attenuation but not selection. Following your suggestion we added a genuine item-level held-out test on the existing logs—fit (a,s,ℓ) on a random half of each config's items, predict slip on the disjoint half (200 splits): held-out slip correlation r=0.95 and held-out slip-MAE 0.07. So the law generalizes across items, not merely refits its own logs. The within-task code slope ≈1, exactly where the judge cannot execute, remains the cleanest evidence for the revision reading.
>
> **5. Proposition 2 (thank you for the counterexample).** You are right; we verified your counterexample holds (inside the Prop. 1 regime the variance can be minimized at K>1). "V(K) non-increasing ⇒ Var non-decreasing" is false: with J decreasing, a shrinking numerator can lower the variance for a step. We removed the claim and now state only the 1/J² variance divergence at the reliability floor (J→0), with K*=1 demoted to a conservative operational default (retrying erodes J; no larger cap improves recovery empirically), not a variance minimizer. The abstract, §4, and the remarks were adjusted to match.
>
> **Strengthening (deployment note).** We added a note that a single binary ruling at K=1 gives a noisier predictor, lowering the judge–gold correlation and shrinking PPI++'s edge toward the gold-only mean (never below it, by the λ* fallback); a few repeated rulings recover most of it, so the recipe degrades gracefully with the ruling/label budget.
>
> Thank you again—these corrections make the paper more precise without touching the flagship results.